

**Tropical atmospheric circulation response to the G1 sunshade geoengineering**
**radiative forcing experiment**
Anboyu Guo[1], John C. Moore[1,2,3], Duoying Ji[1]
[1] College of Global Change and Earth System Science, Beijing Normal University, 19 Xinjiekou
Wai St., Beijing, 100875, China
[2] Arctic Centre, University of Lapland, P.O. Box 122, 96101 Rovaniemi, Finland
[3] CAS Center for Excellence in Tibetan Plateau Earth Sciences, Beijing 100101, China
*Correspondence to:* John C. Moore (john.moore.bnu@gmail.com)
**Abstract.** We investigate the multi-earth system model response of the Walker
circulation and Hadley circulations under the idealized solar radiation management
scenario (G1) and under abrupt4×CO$_2$. The Walker circulation multi-model ensemble
mean shows changes in some regions but no significant change in intensity under G1,
while it shows 4° eastward movement and $1.9×10^9$ kg s$^{-1}$ intensity decrease in
abrupt4×CO$_2$. Variation of the Walker circulation intensity has the same high
correlation with sea surface temperature gradient between eastern and western Pacific
under both G1 and abrupt4×CO$_2$. The Hadley circulation shows significant
differences in behavior between G1 and abrupt4×CO$_2$ with intensity reductions in the
seasonal maximum northern and southern cells under G1 correlated with
equator-ward motion of the Inter Tropical Convergence Zone (ITCZ). Southern and



northern cells have significantly different response, especially under abrupt4×$CO_2$
when impacts on the southern Ferrel cell are particular clear. The southern cell is
about 3% stronger under abrupt4×$CO_2$ in July, August and September than under
piControl, while the northern is reduced by 2% in January, February and March. Both
circulations are reduced under G1. There are good correlations between northern cell
intensity and land temperatures, but not for the southern cell. Changes in the
meridional temperature gradients account for changes in Hadley intensity better than
changes in static stability both in G1 and especially in abrupt4×$CO_2$. The difference in
response to the zonal Walker circulation and the meridional Hadley circulations under
the idealized forcings may be driven by the zonal symmetric relative cooling of the
tropics under G1.
**1 Introduction**
The large-scale tropical atmospheric circulation may be partitioned into two
independent orthogonal overturning convection cells, namely the Hadley circulation
(HC) and the Walker circulation (WC), (Schwendike et al., 2014). The Hadley
circulation is the zonally symmetric meridional circulation with an ascending branch
in the intertropical convergence zone (ITCZ) and a descending branch in the
subtropical zone, and which plays a critical role in producing the tropical and
subtropical climatic zones, especially deserts (Oort and Yienger, 1996). The Walker
circulation is the asymmetric zonal circulation which extends across the entire tropical



Pacific, characterized by an ascending center over the Maritime Continent and
western Pacific, eastward moving air flow in the upper troposphere, a strong
descending center over the eastern Pacific and surface trade winds blowing counter to
the upper winds along the equatorial Pacific completing the circulation (Bjerknes,

46   1969).

Observational evidence shows a poleward expansion of the Hadley circulation in
the past few decades (Hu et al., 2011) and an intensification of the Hadley circulation
in the boreal winter (Song and Zhang, 2007). Moreover, climate model simulations
with increased greenhouse gas forcing also indicate a poleward expansion of the
Hadley circulation, though weaker than that observed (Hu et al., 2013; Ma and Xie,
2013; Kang and Lu, 2012; Davis et al., 2016). Vallis et al. (2015) analysed the
response of 40 CMIP5 climate models finding that there was only modest model
agreement on changes. Robust results were slight expansion and weakening of the
winter cell Hadley circulation in the northern hemisphere. Observational evidence
shows a strengthening and westward movement of the Walker circulation from 1979
to 2012 (Bayr et al., 2014; Ma and Zhou, 2016). However, the time required to
robustly detect and attribute changes in the tropical Pacific Walker Circulation could
be 60 years or more (Tokinaga et al., 2012). Model results suggest a significant
eastward movement with weakening intensity under greenhouse gas forcing (Bayr et
al., 2014).
Geoengineering as a method of mitigating the deleterious effects anthropogenic





climate change has been suggested as a compliment to mitigation and adaptation
efforts. For example, Shepherd et al. (2009) summarized the methodologies and
governance implications as early as a decade ago. Solar radiation management
geoengineering can lessen the effect of global warming due to the increasing
concentrations of greenhouse gases by reducing incoming solar radiation. This
compensating of longwave radiative forcing with shortwave reductions necessarily
leads to non-uniform effects around the globe, as summarized in results for many
climate models in the Geoengineering Model Intercomparison Project (GeoMIP) by
(Kravitz et al., 2013). This is due to the seasonal and diurnal patterns of short wave
forcing being far different from the almost constant long wave radiative absorption. In
addition, solar geoengineering, or solar radiation management (SRM), tends to
produce net drying due to the change in vertical temperature gradient as greenhouse
gasses increase absorption in the troposphere while shortwave radiative forcing
affects surface temperatures (Bala et al., 2011). These differences in short and long
wave forcing impacts atmospheric circulation and hence precipitation patterns,
summarized for the GeoMIP models by Tilmes et al. (2013). There is also a relative
undercooling of the polar regions and overcooling of the tropics and a similar
response of oceans versus land with globally uniform SRM. Extreme precipitation is
affected by SRM such that heavy precipitation events become rarer while small and
moderate events become more frequent (Tilmes et al., 2013). This is generally
opposite to the impact of greenhouse gas forcing alone which tends to produce a "wet
gets wetter and dry gets drier" pattern to global precipitation anomalies (Tilmes et al.,



2013). Finally tropical extreme cyclones have been shown to be affected by
geoengineering in ways that do not simply reflect changes in tropical sea surface
temperatures due to large scale planetary circulations and teleconnection patterns
(Moore et al., 2015).
To date, few studies of the impact of geoengineering on tropical atmospheric
circulation has been published. Ferraro et al. (2014) using an intermediate complexity
climate model found tropical overturning circulation weakens in response to
geoengineering with stratospheric sulfate aerosol injection. But geoengineering
simulated as a simple reduction in total solar irradiance does not capture this effect.
Davis et al. (2016) analyzed 9 GeoMIP models and report that the Hadley circulation
expands in response to a quadrupling of atmospheric carbon dioxide concentrations
more or less proportionality to the climate sensitivity of the climate model, and
shrinks in response to a reduction in solar constant. Smyth et al. (2017) report that
changes in the Hadley cells dominate changes in tropical precipitation under solar
geoengineering, and that seasonal changes mean that the ITCZ has smaller amplitude
migration shifts compared with no geoengineering.
The El Niño Southern Oscillation (ENSO) is the largest mode of multi-annual
variability exhibited by the climate system in terms of its temperature variability and
also for its socio-economic impacts. This tropical circulation pattern is intimately
related to changes in the Walker circulation, and indirectly to the Hadley circulation
by its impacts on global energy balance. Few studies of climate model ENSO





response to geoengineering have been made, with Gabriel and Robock (2015) finding
that stratospheric aerosol injection by the GeoMIP G4 experiment produces no
significant impacts on El Niño/Southern Oscillation. The signal to noise ratio in the
G4 experiment is relatively low with a background of only the modest RCP4.5
greenhouse gas forcing scenario. However, this topic is worthy or more investigation
since one concern is that SRM geoengineering will place the climate system into a
new regime of variability (Robock, 2008; Shepherd, 2009). If this were the case then
we would expect that the dominant climate modes of variability would also differ
from both pre-industrial conditions and those under greenhouse gas forcing alone.
Although this can be studied via volcanic analogues, they are imperfect due to their
transient nature compared with long-term deployment of geoengineering (Robock et
al., 2008). Tropical volcanic eruptions do indeed change the global circulation
(Robock, 2000), and so climate mode change is a potential risk of geoengineering.
Hence examining the tropical circulation and their response under ENSO modulation
can provide evidence on the likelihood of geoengineering inducing a regime change
on the global climate system.

In this paper we utilize simulation results from 8 Earth System Models (ESM)

that participated in the GeoMIP G1 experiment (Kravitz et al., 2011) and compare
these results with the corresponding Climate Model Intercomparison Project Phases 5
(CMIP5) experiment for abrupt quadrupling of $CO_2$ (abrupt4×$CO_2$) and preindustrial
conditions (piControl). The G1 scenario is the largest geoengineering signal addressed



to date by experiments given that it is designed to balance radiative forcing from
quadrupled $CO_2$, hence the signal to noise ratio is high, and furthermore it has been
completed a by a large number of ESM and so we can examine across model
differences in simulations. We address the following key questions: Does the G1
scenario counteract position and intensity variations in the Walker and Hadley
circulations caused by the greenhouse gas long wave forcing under abrupt4×$CO_2$?;
and how does the tropical atmospheric circulation, including the Walker and Hadley
circulations, respond to warm and cold phases of the El Niño Southern Oscillation
(ENSO) in G1 and abrupt4×$CO_2$

**2 Data and methods**
We use 8 ESM (Table 1), a subset of the group described in Kravitz et al. (2013)
that have completed G1. We are limited to these models due to unavailability of some
fields in the output from other models. The simulations in each model are initiated
from a preindustrial conditions which has reached steady state, denoted as piControl,
which is the standard CMIP5 name for this experiment (Taylor et al., 2012). Our
reference simulation, denoted abrupt4×$CO_2$, is also a standard CMIP5 experiment in
which $CO_2$ concentrations are instantaneously quadrupled from the control run. This
experiment implies an atmospheric $CO_2$ concentration of nearly 1140 ppm, close to
"business as usual" scenarios such as RCP8.5 by the year 2100. Experiment G1 in
GeoMIP involves an instantaneous reduction of insolation simultaneous with this $CO_2$



increase such that top-of-atmosphere (TOA) radiation differences between G1 and
piControl are no more than 0.1 W m$^{-2}$ for the first 10 years of the 50 year experiment
(Kravitz et al., 2011). The amount of solar radiation reduction is model dependent but
does not vary during the course of the simulation.

We used the following variables from 8 climate models and reanalysis data (Table

1): sea level pressure (SLP), sea surface temperature (SST), zonal wind ($U$),
meridional wind ($V$) Sea level pressure and sea surface temperature interpolated onto
a regular 1°× 1°grid. The zonal and meridional wind are regridded onto a common
horizontal fixed grid of 2.5°× 2.5°as in many preceding studies (Bayr et al., 2014; Ma
and Zhou, 2016; Stachnik and Schumacher, 2011). All the data we used are
monthly-mean model output. Reanalysis data span the years 1979-2016.

Composite analysis is applied for the study on the influence of ENSO. We follow

Bayr et al. (2014) and use detrended and normalized Nino3.4 index (monthly
averaged sea surface temperature anomaly in the region bounded by 5°N - 5°S, from
170°W - 120°W) as a criteria to select ENSO event. An index > 1 represents an El
Niño event and < -1 a La Niña one (Bayr et al., 2014). We concatenate variables in all
El Niño and La Niña events for each individual model to get El Niño and La Niña
data sets and then calculate ensemble results.



## 2.1 Mass stream-function

The Hadley and Walker circulations represent the meridional and zonal
components of the complete three-dimensional tropical atmospheric circulation. We
follow many previous authors (e.g. Davis et al., 2016; Bayr et al., 2014; Nguyen et al.,
2013; Ma and Zhou, 2016; Yu et al., 2012) in using mass stream-function to
conveniently separate and picture these two convective flows.
The zonal mass stream-function ($\psi_z$) and meridional mass stream-function ($\psi_m$)
are defined as following:
$$\psi_z = \frac{2\pi a}{g} \int_0^{p_s} u_D \, dp \qquad (1) \qquad\qquad \psi_m = \frac{2\pi a \cos(\phi)}{g} \int_0^{p_s} v \, dp \qquad (2)$$

where $u_D$ and $v$ respectively represent the divergent component of the zonal wind and
the zonal-mean meridional wind, $a$ is the radius of Earth, $g$ is the acceleration of
gravity (9.8 ms$^{-2}$), $p$ is the pressure, $p_s$ is the surface pressure, and the $\phi$ in (2) is
latitude. The meridionally averaged $u_D$ between 5°S and 5°N are integrated from top
of the atmosphere to the surface in calculating the zonal mass stream-function ($\psi_z$).
Some previous studies have removed the fast response transient and only use
years 11-50 of G1 and abrupt4×CO$_2$ to avoid climate transient effects (e.g. Smyth et
al., 2017; Kravitz et al., 2013), while Davis et al. (2016) discarded the first 5 years,
noting that the choice is conservative. We examine if zonal and meridional mass
stream-function have transient effects at the start of the simulation. Fig. S1We shows
the time series of the Walker circulation as defined by the vertically averaged value of



the stream function $\psi_z$ (STRF, see section 2.2), and shows that there is variability at
many timescales up to decadal but without significant transient effects. This is
confirmed by statistical analysis of each model; for example there are 4 models
(CCSM4, HadGEM2-ES, IPSL-CM5A-LR and MIROC-ESM) that have significantly
higher STRF in the first 10 years of the abrupt4×$CO_2$ simulation than in following
decades. But this is not due to a transient affecting the first few years, but rather to
higher values around 3 years into the simulation, but this is not unusual for each
model's multiannual and decadal variability. On the other hand, the measures of
circulation that rely on sea surface temperature (Fig. S2) show some difference in the
first decade compared with later periods under abrupt4×$CO_2$. The Hadley cell
vertically averaged stream-function shows similar results and strong seasonal
variability (not shown). Therefore to utilize as much data as possible and increase the
robustness of our statistical analysis, we use all 50 years of G1 and abrupt4×$CO_2$
simulations. We use 100 years of piControl simulations as baseline climate for the
same reason.

**2.2 Walker circulation index**

Four related indices have been used to characterize the Walker circulation

intensity and its position. Tropical Pacific east-west gradients, defined by conditions
in the Darwin region (5°S - 5°N, 80°E - 160°E) and the Tahiti region (5°S - 5°N,
160°W - 80°E) of sea level pressure (ΔSLP) and temperature (ΔSST), (Bayr et al.,



2014; DiNezio et al., 2013; Ma and Zhou, 2016; Vecchi and Soden, 2007; Vecchi et al.,
2006) are highly correlated for all 3 experiments discussed here with $R^2$ around 0.9.
Ma and Zhou (2016) used the vertically averaged value of the stream function $\psi_z$
(STRF), over the western and central Pacific (150°E – 150°W) and this is also very
highly correlated with ΔSST and ΔSLP. As we are interested in the structure of the
circulation, so use either the whole stream function, or the STRF in rest of the paper.
To determine the Walker circulation movement in different experiments, we use
the western edge of Walker circulation to represent it position. The western edge is
defined by the zero value of the vertically averaged $\psi_z$ between 400 – 600 hPa in the
western Pacific 120°E – 180°E, (Ma and Zhou, 2016).

**2.3 Hadley circulation index**
Many authors have separated the northern and southern Hadley circulation cells
simply by dividing by hemisphere (e.g., Davis et al., 2016), but during the active
periods of each cell, the circulation extends across the equator into the opposite
hemisphere. The boundary at the edge of the tropics is also known to change but the
circulation cell rapidly becomes weaker beyond the zero crossing of the rotation sense.
To capture the variability of the Hadley circulation cells we select the season of
maximum intensity for each cell, and measure the strength across its full latitudinal
extent. Thus we define the Hadley circulation intensity for the southern cell with the



meridional stream-function between 40°S and 15°N in July, August and September
(JAS), and the northern cell as the absolute value of mean meridional stream-function
between 15°S and 40°N in January, February and March (JFM). We use the 900 – 100
hPa levels (whereas typically 200 hPa has been the ceiling, (e.g., Nguyen et al., 2013))
to accommodate the raised tropopause under greenhouse gas forcing, while avoiding
boundary effects.

**3 Walker circulation response**
**3.1 Intensity**
The mean state of zonal mass stream-function ($\psi_z$) calculated from 8 ensemble
member mean piControl, ERA-Interim reanalysis and the NCEP2 reanalysis results
are shown in Fig. 1. Zonal mass stream-function ($\psi_z$) can intuitively depict the Walker
circulation which exhibits its strongest convection (positive values) in the equatorial
zone across the Pacific. The Walker circulation center is around 500hPa and 160°W.
Fig. 1 shows that the ensemble piControl circulation has a westward displacement and
the intensity measured by STRF is underestimated by 3% relative to ERA-Interim.
There is a similar structure to the stream function differences between piControl and
NCEP2 reanalysis, but with larger magnitudes than from ERA-Interim.
The relative changes from piControl under G1 and abrupt4×$CO_2$ experiments are
shown in Fig. 2. The features of Walker circulation are very similar in both the G1 and



piControl experiments. In abrupt4×$CO_2$ differences are larger, and include a rise in
vertical extent of the circulation and an eastward shift. This is quantifiably confirmed
by the STRF index increase of just 0.3% in G1 but a decrease of 7% in abrupt4×$CO_2$
relative to piControl, (Table 2). The reanalysis data including ERA-Interim and
NCEP2 respectively show $7.6×10^{10}$ kg s$^{-1}$ and $0.7×10^{10}$ kg s$^{-1}$ stronger intensity. There
is much diversity between individual models (Fig. S3).

**3.2 Position**

The vertically averaged zonal mass stream-function ($\psi_z$) for the ensemble means

of the 3 experiments as a function of longitude are shown in Fig. 3. To quantitatively
measure the position change of the Walker circulation we use the western edge index.
The ERA-Interim and NCEP2 reanalysis data respectively show 10.5° and 18° more
easterly positions than the piControl state. The Walker circulation shifts 0.5°
westward in G1 and 4° eastward in abrupt4×$CO_2$ relative to piControl for the
multi-model ensemble mean. However there is some scatter between models (Table 2).
In the G1 experiment, the Walker circulation strengthens over the western Pacific
around 130°E to 150°E and weakens over the eastern Pacific around 115°W to 80°W,
indicating a westward movement relative to piControl, (Table 2). Thus the pattern is
the opposite of that seen under abrupt4×$CO_2$.

Under G1 there is a westward shift in the ascending branch of the circulation

from about 30°E to about 20°E as indicated by comparing the red shaded region



around 30°E in Fig. 2 with the piControl result in Fig. 1. Fig. S3 shows the anomaly is
present in CanESM2, CCSM4, and NorESM1-M, while 3 models show almost no
change (and indeed are missing the African features in their piControl simulation).
BNU-ESM shows the opposite anomaly while GISS-E2-R shows a complex pattern.
There is only small change in the STRF zero crossing location in the region (Fig. 3)
because of the anomalies are not vertical. This position is at the transition from
tropical West African rainforest to wood and grassland in East Africa under present
climates. The movement westward would impact the rain forests of the Congo basin.
There is no similar positional change under abrupt4×CO$_2$ in the region, though there
are many more changes in the circulation as a whole.

**4 Hadley circulation intensity response**

The climatology of the meridional mass stream-function ($\psi_m$) calculated from
multi-model ensemble mean are shown in Figs. 4 and 5 and the individual models are
shown in Fig. S4. This can intuitive describe the Hadley circulation with a clockwise
rotation in the northern hemisphere and an anticlockwise rotation in the southern
hemisphere. The southern Hadley cell width spans nearly 35°of latitude and the
northern Hadley cell about 25° latitude. The intensity anomalies relative to piControl
from both the reanalysis data sets are less than 19% (Fig. 4).
Circulation anomalies under abrupt4×CO$_2$ (Fig. 5), show increased poleward flow
at upper levels of the troposphere and decreased equatorial flow at lower levels in



both northern and southern Hadley cells. The elevation of the circulation upper
branches rises with increased greenhouse gas concentration, as previously noted
(Vallis et al., 2015), and is likely a consequence of the rise in tropopause height due to
greenhouse gases. The southern cell shows a complex anomaly structure with large
changes also in the Ferrel cell circulation that borders it at higher southern latitudes.
The northern cell anomaly is simple in comparison. Under G1 the changes are largest
near the equatorial margins of the cells, with a clear increase in the strength of the
ascending current. There is no significant change in the upper branch of the
circulation showing that the tropopause is returned to close to piControl conditions
despite the greenhouse concentrations being raised. Seasonal differences illustrate the
changes induced under the experiments in a clearer way than the annual ensemble
result (Fig. 6).

In JAS, when the ITCZ is located furthest north around 15°N, the G1 anomaly

indicates a reduction in the upward branch of the southern cell, or equivalently, a
southern migration of the ITCZ. Similarly in JFM there is a corresponding reduction
in strength of the upwelling branch of the northern cell (Fig. 6). This is a similar result
as obtained by Smyth et al. (2017) who considered the ITCZ position to be defined as
the centroid of precipitation, and found changes in position of fractions of a degree.
Fig. 7 shows that there is a good relationship between the intensity of the southern cell
peak intensity with the motion of the ITCZ, showing that the larger the model
reduction in intensity the more the boundary of the ITCZ moves equatorward. The





correlation for the northern cell is not strong to be significant though still indicates
correlation between intensity and ITCZ position changes. The combined seasonal
effect of both cell changes is a reduced migration of the upwelling branches of the
circulation cells across the equator.

The GISS-E2-R model has strikingly different anomalies under both G1 and

abrupt4×$CO_2$ compared with other models, with much more variability and more
changes in sign of rotation not only within the Hadley cell but in the surrounding
Ferrel cells. If we exclude this model from the ensemble, we get an even clearer result
showing that the movement of the equatorial edge of the Hadley cells (the ITCZ)
totally dominates the response under G1 (Fig. S5).

The situation under abrupt4×$CO_2$ is more complex. There is an increase in

poleward circulation in the upper troposphere in Fig. 5. Similarly there is decrease in
equatorward lower tropospheric flow, though it is apparent that the northern cell
changes are simpler than those in the southern one. Since there is more mass at lower
altitudes the net result is weakening of the circulation cells. The expansion poleward
of the cells can be seen by the blue shading in the lower troposphere around 30°S in
JAS and corresponding red shading around 30°N in JFM. The expansion of the tropics
has been noted both in greenhouse gas simulations and observationally (Davis et al.,
2016; Hu et al., 2011). It is noticeable that the southern expansion appears greater
than the northern one, as was also deduced by Davis et al. (2016) based on the
location of the zero in the vertically integrated stream-function. It is also clear that the



extratropical changes in the Ferrel circulation are more pronounced in the southern
hemisphere than the northern one.
We use the magnitude of the mean southern Hadley cell intensity (as defined in
Section 2.3) during JAS and of northern Hadley intensity during JFM to represent the
model behavior under each climate scenario, and plot differences relative to piControl
in Fig. 8. The multi-model ensemble mean reveals a diminished northern Hadley
intensity under G1 of $-18 \times 10^8$ kg s$^{-1}$ and of $-7 \times 10^8$ kg s$^{-1}$ for abrupt4×CO$_2$. The
southern Hadley intensity in JAS exhibits a fall of $-16 \times 10^8$ kg s$^{-1}$ under G1 but an
increase of $23 \times 10^8$ kg s$^{-1}$ under abrupt4×CO$_2$. The anomalies for most models are
significant, and the ensemble means are hugely significant. The reduction in strength
of the northern hemisphere winter cell was also a robust result of climate models
under RCP8.5, while, in contrast to our Fig. 8 result, the southern cell exhibited
almost no change (Vallis et al., 2015).

**5 ENSO variability of Walker and Hadley circulations**
Many previous study have concluded that the Walker circulation weakens and
shifts eastward during El Niño, with opposite effects under La Niña, (Ma and Zhou,
2016; Power and Kociuba, 2011; Yu et al., 2012; Power and Smith, 2007). While the
Hadley circulation shrinks and strengthens during El Niño and oppositely under La
Niña, (Nguyen et al., 2013; Stachnik and Schumacher, 2011). The G1 solar dimming
geoengineering impacts on the Walker and Hadley circulation during ENSO events



will be discussed in this section.

The Walker circulation difference between G1, abrupt4×CO$_2$ and piControl vary

among models during ENSO events (Fig. S6). But the multi-model ensemble mean
presents a clear picture (Fig. 9). The result show that features of Walker circulation
response to ENSO are significantly changed under abrupt4×CO$_2$ compared with
piControl, while G1 compares quite closely to piControl. Differences between G1 and
piControl only manifest themselves at the eastern (about 165°E-180°E) and western
(about 120°W-90°W) sides of Walker circulation, with a significant westward
movement during El Niño, and no significant changes during La Niña.

In contrast under abrupt4×CO$_2$ almost the whole Walker circulation (about

165°E-105°W) strengthens in intensity and the western edge shifts westward at the
95% statistical significance level during El Niño relative to piControl. During La Niña
there is a significant eastward movement in general.

Hadley circulation responses to ENSO under G1, abrupt4×CO$_2$ and piControl

vary among models (Fig. S7). Fig. 10 shows the ensemble mean results. As with the
Walker circulation, the climatological features of the Hadley cell show more
significant changes under abrupt4×CO$_2$ than G1 compared with piControl.

The most notable feature of Fig. 10 is the increase in intensity during La Niña

between 10°S and 10°N under abrupt4×CO$_2$. This corresponds to changes in the
southern Hadley cell (remembering that the axis of the Hadley cells is northwards of
the equator). Also under the same conditions there is weakening of the northern



Hadley cell between 10°and 20°N. The same features are almost as noticeable for
abrupt4×$CO_2$ for El Niño conditions and hence is a general feature of the
abrupt4×$CO_2$ climate state. Beyond the Hadley cells there are modest, but statistically
significant changes in the Ferrel circulations, particularly in the Southern hemisphere.
Changes under G1 in comparison are much smaller than under abrupt4×$CO_2$, though
there are significant reductions in intensity near the margins of the Hadley cells. The
northern cell is more affected in El Niño, while the southern one more in La Niña
states.

**6 Hadley and Walker circulations relationships with temperature**
**6.1 Walker Circulation**

Changes in tropical Pacific SST dominate the global warming response of the

Walker circulation change (Sandeep et al., 2014). A reduced SST gradient between
eastern and western Pacific drives the weakening of Walker circulation that was seen
in a quadrupled $CO_2$ experiment (Knutson and Manabe, 1995). The temperature
difference between eastern and western Pacific, $\triangle$ SST, explains 96% of the
inter-model variance in the strength of the Walker circulation in the G1-piControl
anomalies and 79% of the variance for abrupt4×$CO_2$-piControl, (Fig. 11). There is no
difference in model behavior between the G1 and abrupt4×$CO_2$ anomalies and $\triangle$SST
explains 83% of the overall variance. Despite a temperature transient of at a decade or
so (e.g. Kravitz et al., 2013) in the abrupt4×$CO_2$ simulation and the lack of any





transient in STRF (Fig. S1), the relationship with ΔSST is nearly as good as for
piControl. This suggests that there is no difference in mode of behavior of the Walker
circulation under solar dimming geoengineering or greenhouse gas forcing, in contrast
with the changes seen in the Hadley cells.

Some models have strong correlation between monthly temperature and Walker

circulation (not shown), with positive correlation in northern hemisphere and negative
correlation in southern hemisphere due to those models having strong seasonality in
their STRF (Fig. S1). The correlation between yearly STRF and global 2 m
temperatures are shown in Fig. 12 and the individual models are shown in Fig. S8. We
discard first 20 years for G1 and abrupt4×CO$_2$ to remove the temperature transients.
In G1 all models except CanESM2 and MIROC-ESM have strong negative
correlations between STRF and tropical Pacific temperatures. BNU-ESM, CCSM4
and NorESM1-M show a positive correlation with temperatures in the South Pacific
convergence zone (SPCZ) and its linear extension in the South Atlantic. These
features are generally muted or absent in the piControl simulations. Experiments
suggest that a key feature of the diagonal structure of the SPCZ is the zonal
temperature gradient in the Pacific which allows warm moist air from the equator into
the SPCZ region. This moisture then intensifies (diagonal) bands of convection
carried by Rossby waves (Van der Wiel et al., 2016). The three models with the
positive correlation between STRF and SPCZ temperatures except BNU-ESM have
increased STRF and ΔSST under G1 (Fig. 11) suggesting that this mechanism is





responsive in at least some of the models to G1 changes in forcing. The SPCZ is the
only part of the ITCZ that extends beyond the tropics and so may be expected to be
more subject to the meridional gradients in radiative forcing produced by G1. The
correlations under abrupt4×$CO_2$ are more variable across the models, though some of
models like IPSL-CM5A-LR, MIROC-ESM and HadGEM2-ES exhibit widespread
anti-correlation between STRF and temperatures; the spatial variability suggests that
this not due to the strong transient response in global temperature rises under
abrupt4×$CO_2$.

**6.2 Hadley Circulation**

We now consider how surface temperature changes may impact the Hadley

circulation. To remove the transients, we only use the last 30 years for G1 and
abrupt4×$CO_2$. The decrease of the northern Hadley cell intensity in JFM (Fig. 8)
correlates with northern hemispheric land temperatures (Fig. 13), explaining 58% of
the variance in model anomaly under G1 – which is nevertheless not significant at the
95% level - and 81% under abrupt4×$CO_2$. Northern hemisphere land temperature also
explains 83% of the G1 anomaly in the southern Hadley cell in JAS, but has no
impact on the abrupt4×$CO_2$ anomaly. We explored the impact of land-ocean
temperature differences by considering the Tibet and tropical ocean temperature
differences (Fig. S9). Results were similar as for Fig. 13, with significant correlations
for G1 in the southern Hadley cell.





Seo et al. (2014) examine the relative importance of changes in meridional
temperature gradients in potential temperature, subtropical tropopause height, and
static stability on the strength of the Hadley circulation. They find that according to
both scaling theory based on the Held and Hou (1980) and the Held (2000) models,
and analysis of 30 CMIP5 models forced by the RCP8.5 scenario, that it is the
meridional temperature gradient that is the most important factor.
We used the same procedure atsSeo et al. (2014) on the 4 models (BNU-ESM,
IPSL-CM5A-LR, HadGEM2-ES, MIROC-ESM) that provide all the fields needed
under G1 and abrupt4×CO2 scenarios (Table 3). The changes in ensemble mean
circulation intensity are similar under G1 and abrupt4×CO2, as are the changes in
potential temperature gradients relative to piControl, but the changes in static stability
are very different between the experiments. The tropospheric heights also change
between G1 and abrupt4×CO2 scenarios, with small reductions under G1 and about a
3% and 0.9% increase respectively in south and north cell under abrupt4×CO2. We
used the two scaling relations given by Seo et al., (2014) to also estimate the change
in Hadley intensity based on the changes in temperature gradients, static stability and
tropospheric height for the ensemble mean of the 4 models (Table 3). Both
formulations give fairly similar numbers for the estimated change in Hadley
intensities in northern and southern cells under G1 and abrupt4×CO2. These estimates
agree with the simulated changes in intensities under G1, but are very different from
those simulated under abrupt4×CO2. The obvious cause of the discrepancies under





abrupt4×CO2 is the change in static stability, which in both model scaling
formulations leads to 18-25% reductions in Hadley intensity compared with the
ensemble model simulated changes of about ±4%. This supports the analysis of Seo et
al. (2014) that it is the meridional temperature gradient that is the dominant factor in
determining the strength of the Hadley circulation.

**7 Discussion**


Our main purpose in this study has been to analyze the response of Walker and
Hadley circulation to greenhouse gas and solar dimming geoengineering forcing
simulated by abrupt4×CO$_2$ and G1 experiments. A clear Walker circulation westward
movement during El Niño and an eastward movement during La Niña are shown
nearly everywhere along the equator in abrupt4×CO$_2$ relative to piControl. However
only the eastern and western side of Walker circulation manifest the same movement
during ENSO events in G1 relative to piControl. The range and amplitudes of
significant changes are smaller in G1 than in abrupt4×CO$_2$. We note a potentially
important change in position of the walker Circulation associated with the West
African rainforest and East African grassland zones, under G1, with potential for the
encroachment of a drier climate into the Congo basin.
Davis et al., (2016) note an expansion in the Hadley cells in proportion to the
temperature rises in the models under both G1 and abrupt4×CO$_2$. Here, we see large
changes throughout the whole Hadley cell circulation under abrupt4×CO$_2$. We also
see that the northern boundary of the Southern cell tends to expand even further



northwards with a corresponding weakening of the northern cell during La Niña
conditions. Global temperatures are relatively reduced during La Niña years. Beyond
the Hadley cells there are modest, but statistically significant changes, in the Ferrel
circulations, particularly in the Southern hemisphere. Changes under G1 in
comparison are much smaller than under abrupt4×$CO_2$, though there are significant
reductions in intensity near the margins of the Hadley cells and these are related to
equator-ward motion of the ITCZ. The northern cell is affected more in El Niño, while
the southern one more by La Niña states.
Davis et al. (2016) show that southern Hadley cell expansion in the tropics is on
average twice the northern Hadley expansion. The idealized forcings in abrupt4×$CO_2$
and G1 show this cannot be due to stratosphere ozone depletion – the mechanism
sometimes used to account for the similar observed greater expansion of the southern
Hadley cell (Waugh et al., 2015). The changes in width of the tropical belt is strongly
dependent on the tropical static stability in the models according to the Held and Hou
(1980) scaling, that is with the potential temperatures at the tropical tropopause (100
hPa) and the surface. Since the adiabatic lapse rates scales with surface temperature,
this is also reflected in the surface temperature. Consideration of simplified
convective systems based on moist static energy fluxes (Davis, 2017), or by making
some assumptions with the Held (2000) and Held and Hou (1980) models led Seo et
al. (2014) to suggest Hadley cell intensity scales according to the equator-pole
temperature gradient.



Furthermore the intensity of the Hadley circulation is expected to decrease as it
expands and also in response to an accelerated hydrological cycle – that is expected
under greenhouse gas forcing, but not solar geoengineering which leads to net drying
(Kravitz et al., 2013). This is cannot be a complete explanation for circulation changes
since the Hadley circulation also depends on the evolution of the baroclinic
instabilities in the extratropics, which may have quite different response to climate
warming (e.g. Vallis et al., 2015). Our analysis of intensity shows differences in
behavior between southern and northern cells, and in particular a lack of a strong
dependences on temperature gradients for the southern cell. The difference in
behavior between northern and southern Hadley cells has not been explained to date.
Seo et al. (2014) note that under RCP8.5 forcing, models of the southern Hadley cell
changes are split almost equally between those predicting increases in intensity and
those that suggest decreases, whereas all but 1 of 30 models predicts a decrease in the
northern cell. We note that the vertical expansion of the circulation under
abrupt4×$CO_2$, has been associated with an expected decrease in the circulation. But
we observe an increase in the southern Hadley cell intensity, while the northern one is
stronger than under the G1 forcing. Our analysis of the relative importance of factors
in driving intensity suggests, as with Seo et al (2014), that the meridional temperature
gradient plays the dominant role rather than tropopause height or static stability
changes.
The response times of the Hadley circulations to changes in radiative forcing are



very fast, as shown by the lack of transients in the simulated time series. Surface
temperature, especially under the strong abrupt4×CO$_2$ forcing takes at least a decade
and parts of the system, such as the ocean and ice, would require even longer to reach
equilibrium. The northern hemisphere continents have faster response times than the
oceans and so we would expect the southern hemisphere to perhaps be much further
from an equilibrium response than the northern one. This is also reflected in the lack
of an equivalent to the "Arctic amplification" seen in the northern hemisphere under
both observed and simulated forcing by greenhouse gases. The lack of anomalous
southern polar warming is linked to the much cooler surface temperatures in the
Antarctic mitigating against both temperature feedbacks and the ice-albedo feedback
mechanism (Pithan and Mauritsen, 2014).
Our analysis of circulation intensity changes and their dependence on temperature
changes shows quite different sets of behavior under G1 than under abrupt4×CO$_2$ for
the Hadley but not the Walker circulation. The response under G1 relative to piControl
is a slight overcooling of the tropics relative to the global mean temperature (Kravitz
et al., 2013). Experiments with idealized climate models (Tandon et al., 2013) show
that heating at the equator alone tends to reduce the Hadley cell width, while wider
heating in an annulus around the outer tropics (20°-35°) tends to produce a complex
response to circulation in both Hadley and Ferrel cells, more reminiscent of the
anomaly patterns seen under abrupt4×CO$_2$. The climate forcing under G1 is designed
to be zonally symmetric, and that may explain lack of impact in the Walker circulation



under both G1 and greenhouse gas forcing. While under the latitudinal varying
forcing of G1 there are clear changes in the Hadley cell. The reduction in incoming
shortwave radiation in G1 would intuitively mean reduced heating and moisture flux
in the ITCZ, which follows the movement of the sun. Reduced ocean heating would
then tend to mean a smaller amplitude of seasonal movement of the ITCZ. Analysis of
extreme precipitation events in daily data from the GeoMIP models (Ji et al.,
submitted to ACP) shows that the frequency of the Rx5day extreme is decreased
under G1 along a seasonal path that follows the ITCZ motion, while precipitation
extremes increase in the tropical dry seasons. This result is consistent with the
variation in the Hadley intensity cell seen here.
Both models and the limited observational data available on the Hadley
circulation indicate that it is not zonally symmetric: there are intense regions at the
eastern sides of the oceanic basins (Amaya et al., 2017), and much of the natural
variability of the circulation is related to ENSO. This and the opposite correlations
with surface temperatures in the Pacific and SPCZ with STRF under G1 (Fig. 12)
suggests an interplay between Hadley and Walker circulations that could repay further
consideration of model data at seasonal scales. The importance of the tropical ocean
basins as genesis regions for intense storms also suggests that changed radiative
forcing there under geoengineering could cause important differences in seasonal
precipitation extremes, that maybe hidden in monthly or annual datasets.



*Acknowledgements.* We thank the climate modeling groups for participating in the
Geoengineering Model Intercomparison Project and their model development teams;
the CLIVAR/WCRP Working Group on Coupled Modeling for endorsing the GeoMIP;
and the scientists managing the earth system grid data nodes who have assisted with
making GeoMIP output available. This research was funded by the National Basic
Research Program of China (Grant 2015CB953600).

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

FIGURES



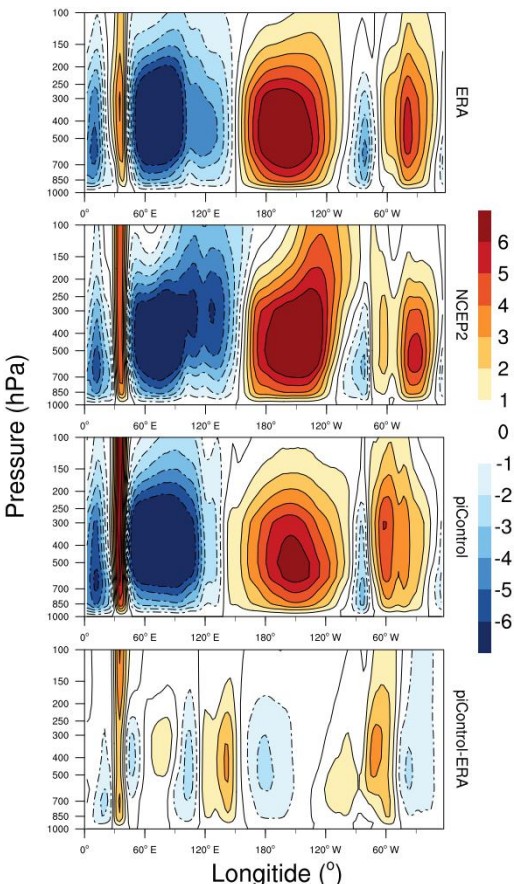


**Figure 1.** The ERA-Interim reanalysis (top), NCEP2 reanalysis (second row), model

ensemble mean Walker circulation under piControl (third row) and difference between

ERA-Interim and piControl (bottom). Color bar indicates the value of averaged zonal

mass stream-function ($10^{10}$ kg s$^{-1}$). Warm color (positive values) indicate a clockwise

rotation and cold color (negative values) indicate an anticlockwise rotation.




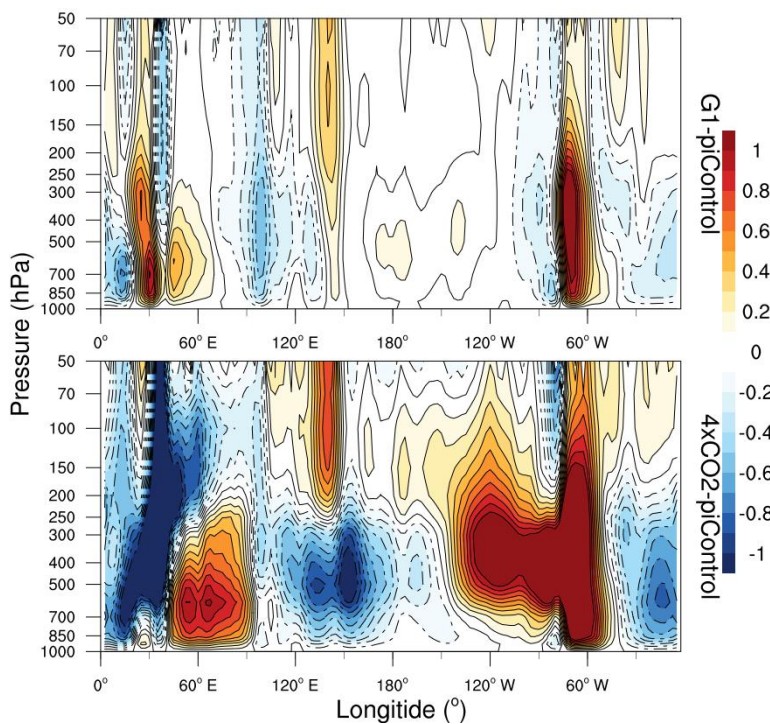


**Figure 2.** Same as Fig. 1. But the top and bottom respectively indicate the anomalies

relative to piControl for G1 and abrupt4×CO$_2$ experiments.



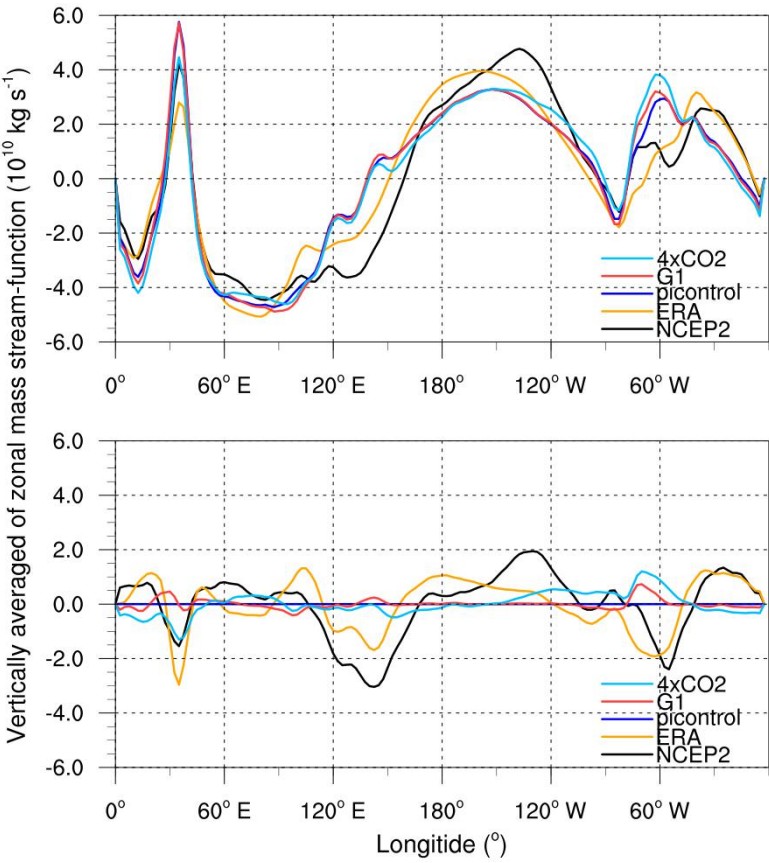

**Figure 3.** The vertically averaged zonal mass stream-function ($10^{10}$ kg s$^{-1}$) in piControl, G1, abrupt4×CO2 experiment for ensemble mean, ERA-Interim and NCEP2 are in the top panel. Lines in bottom panel are the difference between piControl and other scenarios.



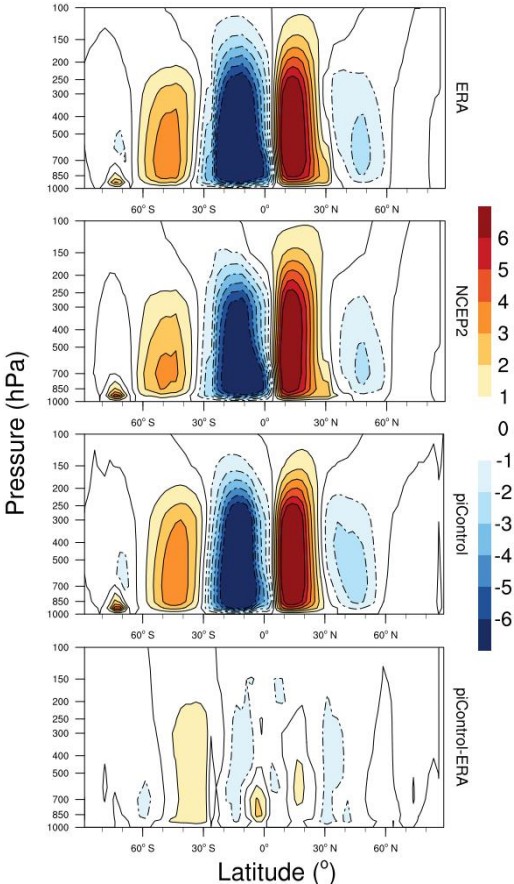

**Figure 4.** The ERA-Interim reanalysis (top), NCEP2 reanalysis (second row), model

ensemble mean Hadley circulation under piControl (third row) and difference

between ERA-Interim and piControl (bottom). Color bar indicates the value of

averaged meridional mass stream-function ($10^{10}\,\text{kg s}^{-1}$). Warm colors (positive values)

indicate a clockwise rotation and cold colors (negative values) indicate an

anticlockwise rotation.





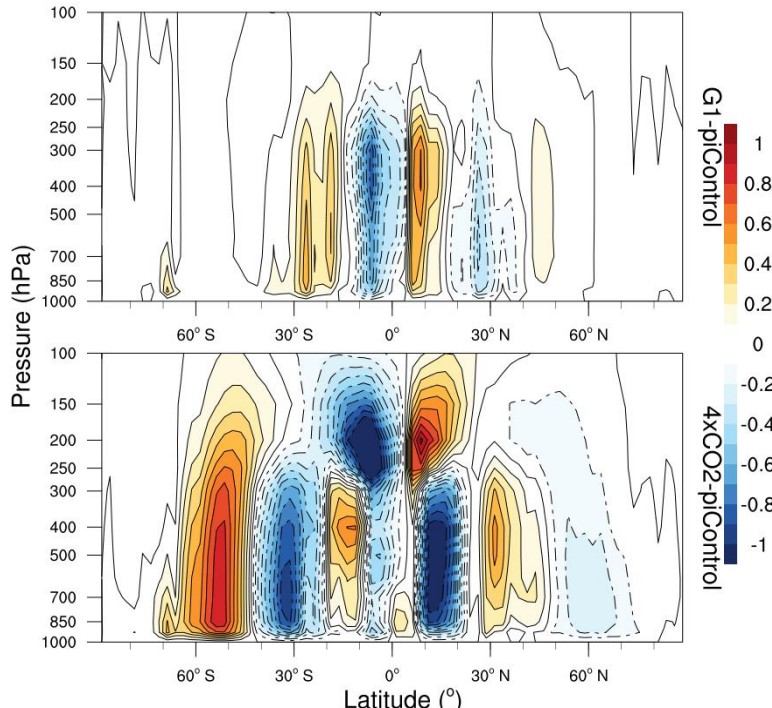


**Figure 5.** Model ensemble mean meridional stream-function anomalies G1-piControl

(top) and abrupt4×CO₂-piControl (bottom). Contours and color bar indicate the value

of averaged meridional mass stream-function ($10^{10}$kg s$^{-1}$).





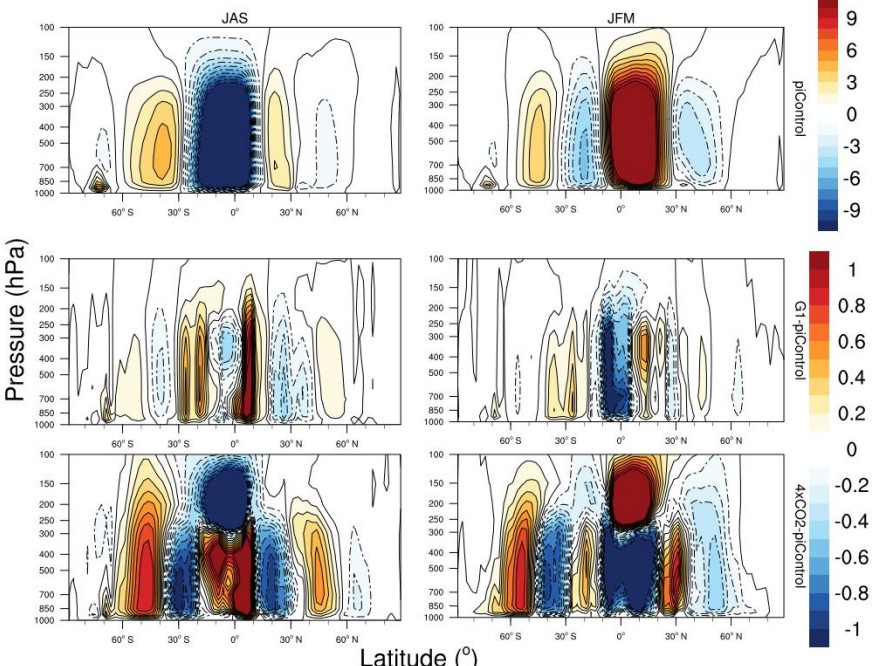

**Figure 6.** Model ensemble mean meridional stream-function in JAS (left) and JFM

(right). Top shows piControl, while center and bottom row respectively indicate the

anomalies relative to piControl for G1 and abrupt4×CO$_2$ experiments. Color bar

indicates the value of averaged meridional mass stream-function ($10^{10}$ kg s$^{-1}$). Warm

colors (positive values) indicate a clockwise rotation and cold colors (negative values)

indicate an anticlockwise rotation.





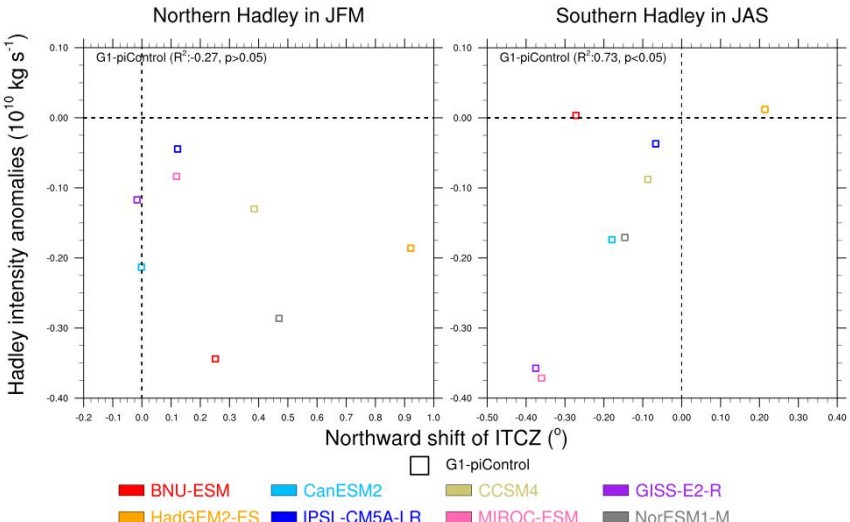

**Figure 7.** Change of Hadley cell intensity as a function of ITCZ position under G1 relative to piControl across the models. The ITCZ position is defined from the centroid of precipitation (Smyth et al., 2017).



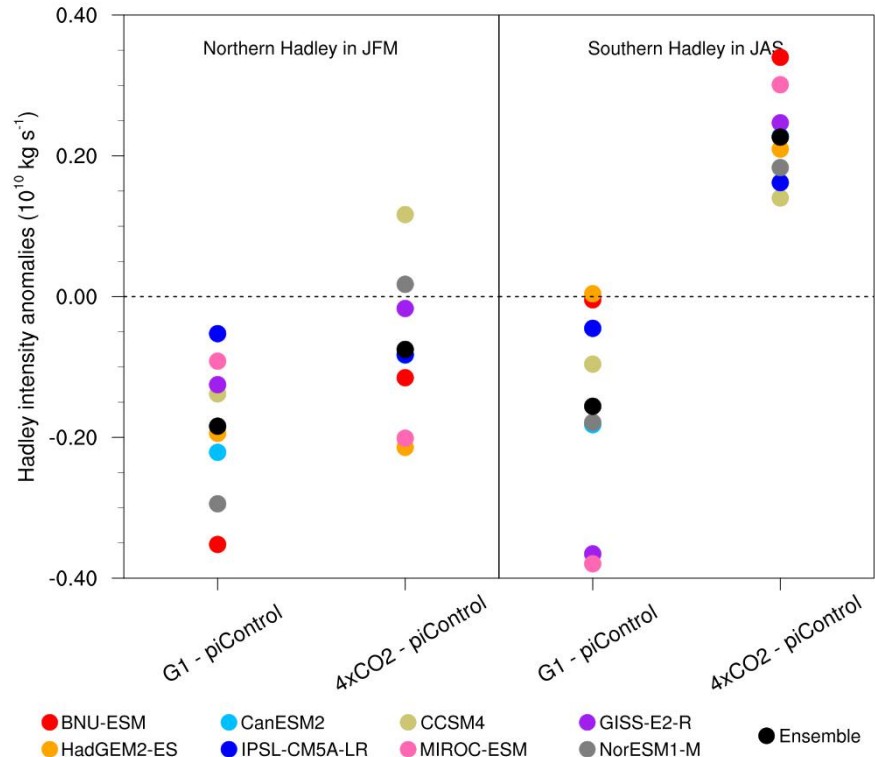

**Figure 8.** Anomalies ($10^{10}$ kg s$^{-1}$) amongst models in Hadley circulation for the

southern cell in JAS (left panel), defined as the magnitude of the mean meridional

stream-function between 15°N and 40°S, and (right panel) the northern cell in JFM,

defined as the magnitude of the mean meridional stream-function between 15°S and

40°N. The dot size for the models is about 1 standard error of the model mean.

804





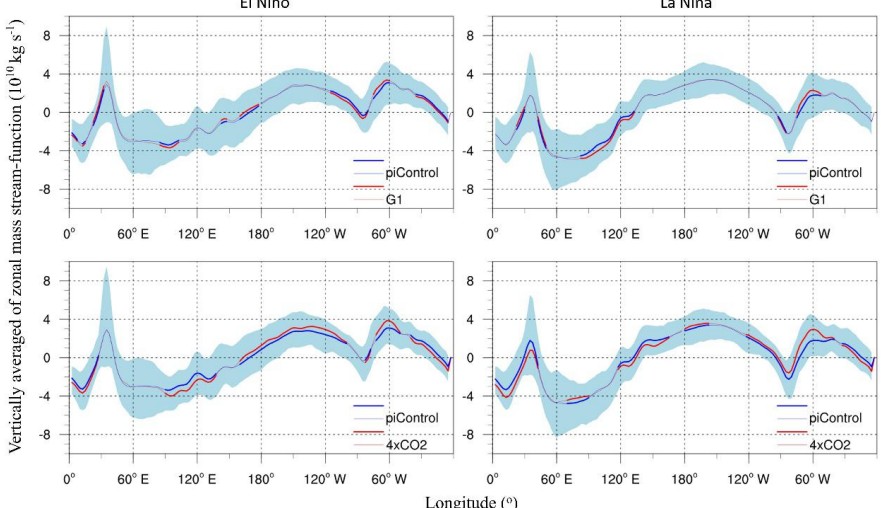

**Figure 9.** The vertically averaged of zonal mass stream-function under ENSO. For El
Niño or La Niña conditions, blue line in each panel represent the vertically averaged
of zonal mass stream-function ($10^{10}$ kg s$^{-1}$) under piControl. Red line in top row is G1
and bottom row abrupt4×CO2. Thick lines denote locations where circulation changes
are significant at the 95% confidence level. The 16%-84% range across the 8
individual models are show by light blue shading.





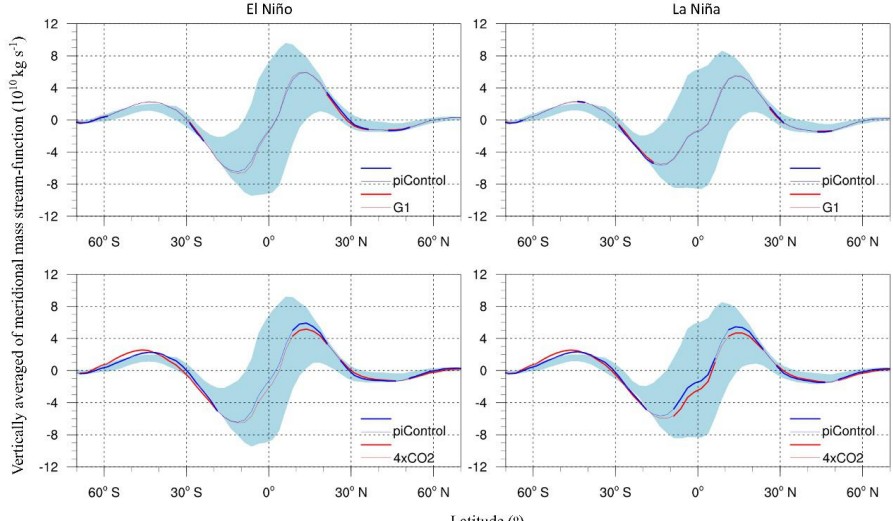

813

**Figure 10.** The vertically averaged of meridional mass stream-function under ENSO.

For El Niño or La Niña conditions, blue line in each panel represent the vertically

averaged of zonal mass stream-function ($10^{10}$ kg s$^{-1}$) under piControl. Red line in top

row is G1 and bottom row abrupt4×CO$_2$. Thick lines denote locations where

circulation changes are significant at the 95% confidence level. The 16%-84% range

across the 8 individual models are show by light blue shading.

820





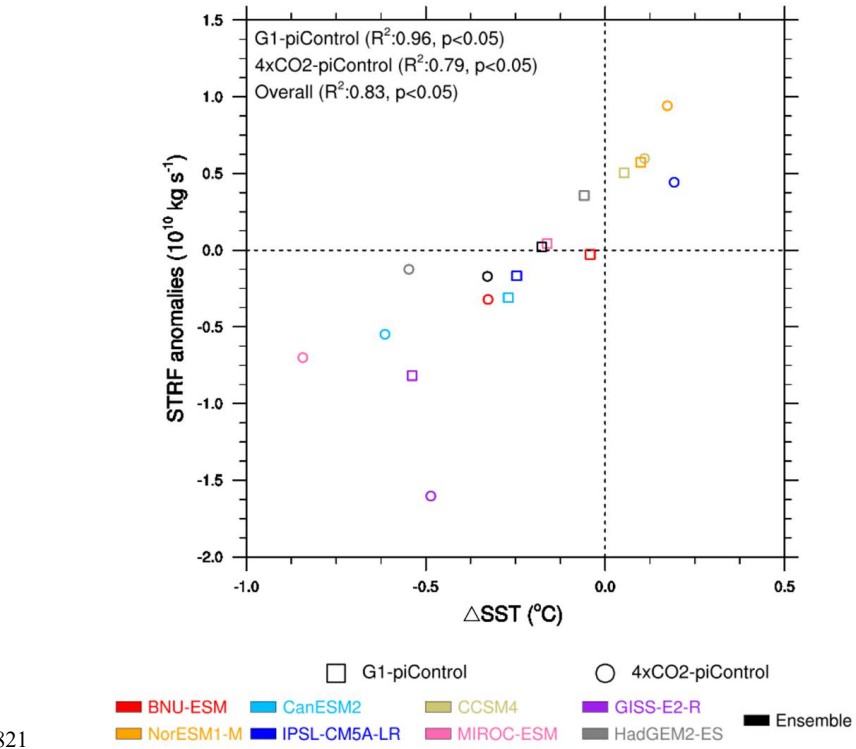

**Figure 11.** Model mean monthly anomalies relative to each model's piControl of STRF and ΔSST. Positive value of STRF and ΔSST indicate strengthening of the Walker circulation.





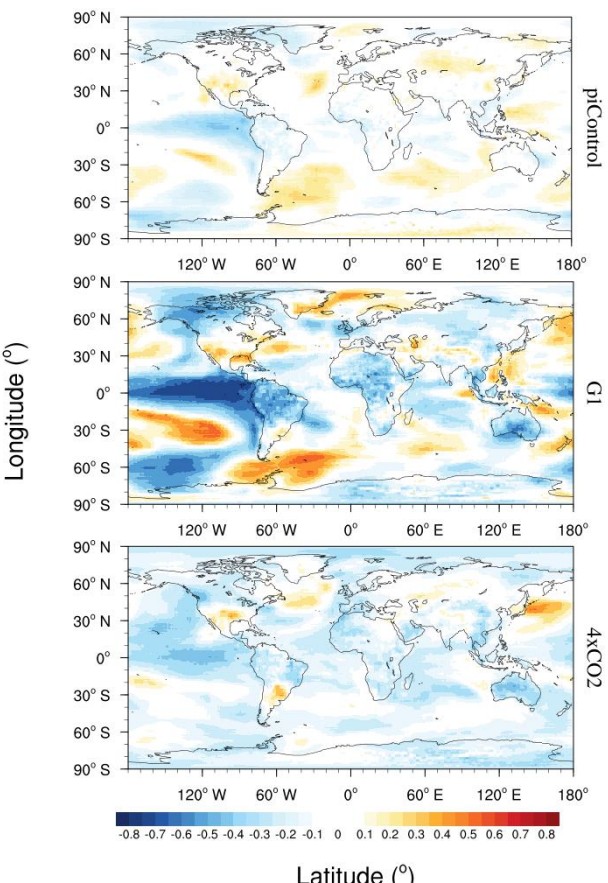

826

**Figure 12.** Mean correlation between yearly STRF and global gridded 2 m
temperatures for 100 years of piControl (top row), and the final 30 years of G1
(middle row) and abrupt4×CO$_2$ (bottom row) experiments for 8 models ensemble
mean.

831




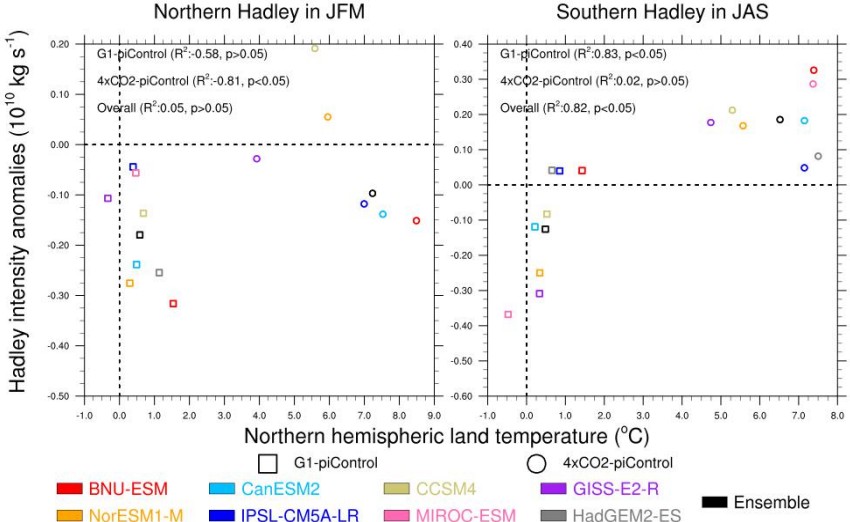

**Figure 13.** Hadley intensity mean model anomalies versus the northern hemisphere land temperature for the northern Hadley cell (left) in JFM and the southern Hadley cell in JAS (right). Positive value of Hadley intensity indicates Hadley circulation strengthening regardless of the direction.

**Table 1.** The GeoMIP, CMIP5 models and reanalysis data used in the paper

| No. | Model[1] | Reference | Lat × Lon |
| --- | --- | --- | --- |
| 1 | BNU-ESM | Ji et al. (2014) | 2.8°×2.8° |
| 2 | CanESM2 | Arora et al. (2011) | 2.8°×2.8° |
| 3 | CCSM4 | Gent et al. (2011) | 0.9°×1.25° |
| 4 | GISS-E2-R | Schmidt et al. (2014) | 2°×2.5° |



| 5 | HadGEM2-ES | Collins et al. (2011) | 1.25°×1.875° |
| 6 | IPSL-CM5A-LR | Dufresne et al. (2013) | 2.5°×3.75° |
| 7 | MIROC-ESM | Watanabe et al. (2011) | 2.8°×2.8° |
| 8 | NorESM1-M | Bentsen et al. (2013), Iversen et al. (2013) | 1.9°×2.5° |
| 9 | NCEP-DOE (NCEP2) | Kanamitsu et al. (2002) | 2.5°×2.5° |
| 10 | ERA-Interim | Simmons et al. (2007) | 0.75°×0.75° |

**1. Full Names:** BNU-ESM, Beijing Normal University-Earth System Model; CanESM2, The
Second Generation Canadian Earth System Model; CCSM4, The Community Climate System
Model Version 4; GISS-E2-R, Goddard Institute for Space Studies ModelE version 2;
IPSL-CM5A-LR, Institut Pierre Simon Laplace ESM; MIROC-ESM, Model for Interdisciplinary
Research on Climate-Earth System Model; NorESM1-M, Norwegian ESM.

**Table 2.** The change of Walker circulation position (°) and intensity ($10^{10}$ kg s$^{-1}$) in 8
models and their ensemble mean. The number in the brackets represent percentage
change relative to piControl. Negative position (STRF) represent westward movement
(weakening) and positive value represent eastward movement (strengthening).
Statistically significant differences at the 5% are in shown in bold.

| Earth System Model | G1 | abrupt4×CO$_2$ |
| --- | --- | --- |





|  | Position | STRF | Position | STRF |
| --- | --- | --- | --- | --- |
| BNU-ESM | 0.32(0.2) | -0.04(-2.3) | **8.6(5.8)** | **-0.34(-18)** |
| CanESM2 | **3.8(2.7)** | **-0.32(-11)** | **16.4(11.5)** | **-0.56(-19.3)** |
| CCSM4 | -1(-0.7) | **0.5(20.6)** | -0.3(-0.2) | **0.58(24.6)** |
| GISS-E2-R | **10.6(6.5)** | **-0.83(-73.5)** | **21.2(13)** | **-1.6(-142.7)** |
| HadGEM2-ES | -1(-0.7) | **0.34(10.8)** | **4.9(3.3)** | **-0.14(-4.4)** |
| IPSL-CM5A-LR | 1.4(1) | **-1.8(-7.8)** | 0.15(0.1) | **0.43(18.3)** |
| MIROC-ESM | 0.5(0.4) | 0.03(0.7) | **-5.2(-4.2)** | **-0.72(-19.1)** |
| NorESM1-M | **-3.7(-2.3)** | **0.56(20.2)** | **-6.6(-4.2)** | **0.93(33.6)** |
| Ensemble | -0.5(-0.3) | 0.007(0.3) | **4(2.8)** | **-0.19(-7.1)** |


**Table 3.** The percentage changes in G1-piControl and abrupt4×CO$_2$-piControl relative
to piControl in a 4 model (BNU-ESM, IPSL-CM5A-LR, HadGEM2-ES,
MIROC-ESM) ensemble mean. Functions 1 and 2 are scale factors for Hadley
circulation (Seo et al., 2014). Function 1 is $\frac{5}{2}\frac{\delta H}{H} + \frac{5}{2}\frac{\delta \Delta_H}{\Delta_H} - \frac{\delta \Delta_V}{\Delta_V}$ and is based the model
of Held and Hou (1980), while function 2 is $\frac{9}{4}\frac{\delta H}{H} + 2\frac{\delta \Delta_H}{\Delta_H} - \frac{3}{4}\frac{\delta \Delta_V}{\Delta_V}$ which is derived
from the model by Held (2000). $\Delta_H$ is meridional temperature gradient defined as
$\frac{\theta_{eq} - \theta_{higher\,lat}}{\theta_0}$ which is the tropospheric mean meridional potential temperature gradient
with $\theta_0$ denoting the hemispheric troposphere mean potential temperature and $\theta_{eq}$



calculated between 10°N and 10°S. We follow Seo et al. (2014) in taking $\theta_{higher\ lat}$ as
the average potential temperature between $10°\text{-}50°$N for the northern hemisphere
winter and $10°\text{-}30°$S for the southern hemisphere. $\Delta_V = \frac{\theta_{300}-\theta_{925}}{\theta_0}$ is the dry static
stability of the tropical troposphere. $H$ is the tropical tropopause height estimated as
the level where the lapse rate decreases to 2°C km⁻¹. The Hadley intensity $\psi_m$ is
described in section 2.3.

| Scenario | G1-piControl | | abrupt4×CO₂-piControl | |
|---|---|---|---|---|
| | North | South | North | South |
| Temperature gradient | -2.6 | -1.2 | -4.4 | -4 |
| Static stability | -3.4 | -3.2 | 21 | 23 |
| Subtropical tropopause height | -0.1 | -0.5 | 0.87 | 3 |
| Function 1 | -3.35 | -1.05 | -29.8 | -25.5 |
| Function 2 | -2.9 | -1.13 | -22.6 | -18.5 |
| Hadley intensity | -3.7 | -1.2 | -3.4 | 4.3 |
