# Peer review of "Tropical atmospheric circulation response to the G1 sunshade geoengineering"

_Atmospheric Chemistry and Physics, 2018_

## Referee Comment (RC1) · Anonymous Referee #1 · 15 Mar 2018

This manuscript is a valuable contribution to the geoengineering literature, as it provides a detailed assessment of the simulated tropical circulation response to uniform solar dimming in a suite of coupled climate models. The Hadley circulation does not return to preindustrial conditions in a climate with quadrupled carbon dioxide levels and reduced insolation (the G1 experiment). The authors attribute this result to changes in meridional temperature gradients rather than changes in static stability. The Walker circulation, by contrast, is largely restored to its preindustrial state in G1. The Introduction section effectively describes the many motivations for the study.

The novelty of this work lies in its assessment of the G1 experiment, as much anal-

ysis of the impact of elevated atmospheric carbon dioxide concentrations on tropical dynamics has been discussed previously (see references below). At present, the Introduction mentions a few studies on the latter subject (on page 3), but a more thorough review of the existing literature is warranted, both in the Introduction and the Discussion. This manuscript would be more effective if it were shortened so as to emphasize new knowledge; if the circulation changes in abrupt4xCO2 simulations differ from what is reported in the existing literature on the subject, this can be emphasized, but otherwise the geoengineering results should be brought to the forefront.

I have questions pertaining to the methodological choices described in Section 2.3. Why is the Hadley cell intensity based on such a broad latitudinal extent (to 40° S or N)? This extends beyond the tropics and includes the Ferrel cell. Additionally, the Hadley cell migration is not symmetric in the two seasons (the July-September cell extends further into the summer hemisphere than the January-March cell), so why is the Hadley cell intensity metric hemispherically symmetric?

Finally, throughout the paper it would be helpful to explicitly distinguish between robust results and results for which the ensemble mean is dominated by inter-model cancellation. For example, it is important to elaborate on this last sentence of Section 3.1 (line 253). Does the substantial inter-model spread undermine subsequent interpretation of the ensemble mean change?

SPECIFIC COMMENTS

The ENSO-related results could be included in the abstract.

Throughout the paper, starting in the abstract, the Hadley circulation changes are discussed in terms of "seasonal maximum northern and southern cells." It would be clearer to discuss these together as the "solstitial Hadley cells," or as "the JAS and JFM cross-equatorial cells."

(lines 79-80) It is not entirely clear what is meant by this phrase: "and a similar response

of oceans versus land."

(line 84) Held and Soden (2006) are better known for explaining this P-E scaling, which is derived from the Clausius-Clapeyron relation and only valid over ocean (reference below).

(lines 108-110) This sentence is unclear: "The signal to noise ratio [. . .]."

(line 198-200) Are all 50 years used only for those measures that do not rely on sea surface temperature?

(line 213) Does the phrase "whole streamfunction" mean for all longitudes? Say that explicitly.

(line 237) Does mean state refer to annual mean state?

Labeling multi-panel figures would facilitate the discussion of results in the text.

(line 308) What constitutes a "good relationship?"

(line 312-314) Note that this was reported by Smyth et al. (2017).

(lines 321-344) This section can be made more concise by focusing on similarities or differences from previous studies on the subject (cf. major comment above).

(lines 339-340) Cite other studies which have noted that solar dimming results in an overcompensation of tropical circulation changes induced by global warming.

You might consider discussing all of the analysis of the Walker and Hadley circulation responses together, i.e. moving current Section 6.1 to follow current Section 3.2.

(lines 399-403) This section is confusing. "Monthly temperatures" in which region?

(lines 433-436) Why do you choose to analyze temperatures over Tibet if the general land-ocean temperature contrast is of interest? This choice should be justified, or the analysis modified. Are you considering surface temperatures? The methodology is not described in enough detail for the results to be reproduced. Additionally, there is an

extensive body of literature connecting inter-hemispheric temperatures and the Hadley circulation/Intertropical Convergence Zone, such as Broccoli et al. (2006), which can be referred to here.

(lines 445-448) Specify the location and season on which this is based. Are these near-surface potential temperature gradients?

(lines 468-470) The sentence "However only the eastern and western [...]" is confusing.

The analysis of the Walker circulation should be better framed. For example, He and Soden (2015) explain that weakening of the Walker circulation due to carbon dioxide forcing is mostly driven by the change in mean sea surface temperatures (SST).

(line 500) The second half of this sentence is unclear.

(lines 514-516) Rephrase this sentence for clarity: "But we observe [...] G1 forcing."

(line 545) Does "ocean heating" refer to warming sea surface temperatures or ocean heat uptake?

(line 548) Define "Rx5day extreme"

Figure 10: Is this based on annual data from El Nino years, or data from a particular season? In a few places in the paper it is not immediately clear what averaging periods and spatial domains are used for calculations.

TECHNICAL COMMENTS

(Line 10) capitalize "Earth"

(Line 25) "good correlations" should be rephrased more precisely

(Line 55) typically capitalized "Northern Hemisphere"

(line 29) "response to" should say "responses of"

(Line 63) "compliment" should be "complement"

(line 157) not a sentence

(line 349-351) This is not a sentence.

(line 542) "While under [. . .]" is not a sentence.

REFERENCES

*A sampling of the many studies examining the Hadley cell response to global warming:

Frierson, D. M. W. (2006). Robust increases in midlatitude static stability in simulations of global warming, Geophys. Res. Lett., 33, L24816, doi:10.1029/2006GL027504

Frierson, D. M. W., J. Lu, and G. Chen (2007). Width of the Hadley cell in simple and comprehensive general circulation models, Geophys. Res. Lett., 34, L18804, doi: 10.1029/2007GL031115.

Held, I.M. and B.J. Soden (2006). Robust Responses of the Hydrological Cycle to Global Warming. J. Climate, 19, 5686–5699, https://doi.org/10.1175/JCLI3990.1

Johanson, C.M. and Q. Fu (2009). Hadley Cell Widening: Model Simulations versus Observations. J. Climate, 22, 2713–2725, https://doi.org/10.1175/2008JCLI2620.1

Lau, W. K., & Kim, K. M. (2015). Robust Hadley Circulation changes and increasing global dryness due to CO2 warming from CMIP5 model projections. Proceedings of the National Academy of Sciences, 112(12), 3630-3635.

Lu, J., G. A. Vecchi, and T. Reichler (2007). Expansion of the Hadley cell under global warming, Geophys. Res. Lett., 34, L06805, doi: 10.1029/2006GL028443.

Seager, R., N. Naik, and G.A. Vecchi (2010). Thermodynamic and Dynamic Mechanisms for Large-Scale Changes in the Hydrological Cycle in Response to Global Warming. J. Climate, 23, 4651–4668, https://doi.org/10.1175/2010JCLI3655.1

Seidel, D. J., Fu, Q., Randel, W. J., & Reichler, T. J. (2008). Widening

of the tropical belt in a changing climate. Nature Geoscience, 1(1), 21-24. doi:http://dx.doi.org.ezproxy.princeton.edu/10.1038/ngeo.2007.38

Additional References

Broccoli, A. J., K. A. Dahl, and R. J. Stouffer (2006). Response of the ITCZ to Northern Hemisphere cooling, Geophys. Res. Lett., 33, L01702, doi: 10.1029/2005GL024546.

He, J., & Soden, B. J. (2015). Anthropogenic weakening of the tropical circulation: The relative roles of direct CO2 forcing and sea surface temperature change. Journal of Climate, 28(22), 8728-8742.

---

## Referee Comment (RC2) · Anonymous Referee #2 · 20 Mar 2018

The authors examine the response of the combined Hadley-Walker circulation to idealized greenhouse gas forcing and solar geoengineering experiments. Solar geoengineering counters most of the changes in the Walker circulation in response to quadrupled carbon dioxide. Weakening sea surface temperature gradients are associated with a weakening Walker circulation across experiments and models. In contrast, changes in surface temperature and zonal temperature gradients do not effectively predict changes in Hadley circulation intensity, some behavior differs in the two hemispheres, and Hadley circulation intensity remains reduced under geoengineering. The authors hypothesize this may be due to changes in meridional temperature gradients, which project more strongly onto the Hadley circulation than onto the Walker circula-

tion.

This manuscript builds upon previous work and presents new insights into Walker-Hadley circulation change under geoengineering and greenhouse gas forcings. I think the manuscript is well-structured, the conclusions follow from the results, and the authors have done sufficient work to warrant publication. I have some suggestions for the text that would improve its clarity, suggestions for relevant literature that the authors might want to consider, and some suggestions on the figures. I think addressing these would elevate the manuscript and help it reach a broader audience.

=====General Comments=====

While the introduction has sufficient breadth, there are areas where it lacks a discussion of more recent work. For example, regarding the statement "...climate model simulations...indicate a poleward expansion of the Hadley circulation, though weaker than that observed", there numerous studies suggesting that this may not be the case. Choi et al. 2014 and Quan et al. 2014 both suggest that reanalysis trends in the HC edges may be overstated, especially compared to independent observations. And it appears that the model trends may not be so different from the reanalysis trends (Garfinkel et al. 2015, Davis and Birner 2017). The choice of metric also matters (Solomon et al. 2016). My understand is that it's actually unclear whether models, reanalyses, and observations disagree on the response of the Hadley circulation; but that itself is a valid motivation. The authors may also consider connecting their work to studies like Schmidt and Grise (2016), who have investigated some of the longitudinal characteristics of Hadley cell variability. Full references can be found at the end.

I think the GISS-E2-R model should be excluded from the composite figures, while the composites with GISS should be shown in the supplementary information (opposite to what is currently done). The authors have made a good choice to show composites with and without GISS, as its behavior deviates so much from the other models, but I think swapping these figures would better support their conclusions and interpretations

while still maintaining full disclosure.

It would be helpful if the authors were more explicit and definitive throughout the manuscript. For example, on lines 97-100, it would be helpful to readers if the sign of the changes were stated, e.g., "...report that decreases in Hadley cell intensity drive the reduction in tropical precipitation under solar geoengineering...". Or, on lines 144-146, state whether the abrupt4xCO2 experiment is close to RCP8.5 in terms of CO2 ppm or in terms of radiative forcing. What follows is a list of some but not all of these instances: -Line 74 -Line 80 -Lines 103-104 -Lines 223-224 -Lines 251-252 -Lines 293-294 -Lines 337-340 (are intensity changes signed, or in terms of absolute value?) -Lines 480-482

I have difficulty discerning information from the anomaly contour figures, like Fig. 2. Standard practice is to show control values overlaid as contours on the shading, so that shifts/expansions/contractions can be more easily discerned. This is especially critical for the Walker circulation - its mean structure and response have substantial spatial variability.

The questions posed at the end of the introduction are a great way to orient the reader. It may be worth specifically restating these questions in the discussion as a way of summarizing the results.

The figure production quality is high, but the image quality is low. Per ACP guidelines, PDF or EPS is preferred so that the figures are crisp when zoomed in (save with vector graphics enabled; in MATLAB, it's the "painters" renderer, not sure how this works in IDL or other languages). Otherwise, I think the DPI needs to be increased for the figures.

=====Specific Comments=====

Line 65: Define "SRM" here rather than on line 73.

Line 99: What are the seasonal changes?

Line 135: ENSO was already previously defined.

Lines 191-194: What is the order of the variability in the first 3 years - 1 sigma, 2 sigma? This may help convince readers it's nothing to worry about.

Lines 227-233: I am somewhat unclear on the metrics for Hadley cell intensity. It seems like the authors are using the average of the 900-100 hPa stream function; are they using a point maximum or an area average?

Line 239, 283-285: "Intuitively": "effectively" or "naturally" might be more appropriate?

Line 243: Mentioning a specific number is good, as is describing how it is derived.

Lines 263-266: The scatter here is very large among the models, even neglecting GISS, which I think is worth noting.

Lines 289-290: It might be more effective to say that there is enhanced overturning aloft and weakened overturning at lower levels; as written, it almost sounds like the anomalies don't conserve mass (reduced equatorward + enhanced poleward).

Line 309: How is the ITCZ metric defined?

Lines 321-325: I think this was essentially said previously on lines 288-294.

Line 333: Could write simply "more pronounced in the southern hemisphere", as the "than…" is implied.

Line 341: I think this is too colloquial, maybe state the p-value or % significance it reaches - stating something like "99.9% significant" is more convincing than "hugely".

Lines 399-400: What is meant by "monthly temperature"?

Line 409: Which experiments? Citation?

Lines 413-415: Suggest stating model names, or at least specifying "two models" instead of "three…except BNU".

Lines 420-421: But isn't it the case that there are still some weak, positive correlations in regions like the SPCZ?

Section 6.2/Table 3: I suggest mentioning the highest and lowest model value for each category, or really anything to help illustrate the model spread. With an N of 4, the average doesn't mean as much, but I think this section is still worth including. For the critical relationships, it may help to show them in scatter plots. I'm curious what output fields are needed that are lacking in most of the models?

Lines 513-514: Why is this expected? Does it follow from the vertical expansion, or from the Held and Soden static stability/Clausius-Clapeyron scaling?

Lines 520-524: Grise and Polvani (2016) would be a good reference for the dynamical response in abrupt4xCO2 outpacing the thermodynamic response. Doesn't this call into question the importance of static stability and meridional gradients in driving the changes in the circulation, if the circulation responds faster? Is it possible that the thermodynamic responses the authors examine might follow from some of the circulation changes, as these circulations transport heat?

Line 548: What is "Rx5day"? I would rewrite this more generally, and avoid acronyms unless they will be useful later.

Lines 553-554: "Intense" - subsidence?

Figure 1 caption: Suggest rewriting so description doesn't only apply to the third subplot, i.e., "Walker circulation in the ERA-Interim reanalysis (top), . . . , model ensemble mean under piControl (third row), . . .".

Figures 7, 8, 11, 13: I think the authors have crafted the color scheme to avoid some of the major color-blindness combinations (i.e., no green/red), but a further helpful step is to not rely solely on color when trying to distinguish data points. I encourage the authors to use different symbols/shapes, in addition to their current color scheme, if they want to communicate values for each model individually rather than the behavior

of the models as a whole.

Table 2: How is the percent change in position defined?

Table 3: The definitions should not be in the caption, but should instead be in the text.

=====References=====

Choi, J., S.-W. Son, J. Lu, and S.-K. Min (2014), Further observational evidence of Hadley cell widening in the Southern Hemisphere, Geophys. Res. Lett., 41, 2590-2597.

Davis, N. and T. Birner (2017), On the Discrepancies in Tropical Belt Expansion between Reanalyses and Climate Models and among Tropical Belt Width Metrics, J. Climate, 30, 1211–1231

Garfinkel, C. I., D. W. Waugh, and L. M. Polvani (2015), Recent Hadley cell expansion: The role of internal atmospheric variability in reconciling modeled and observed trends, Geophys. Res. Lett., 42, 10,824–10,831.

Grise, K. M., and L. M. Polvani (2016), Is climate sensitivity related to dynamical sensitivity?, J. Geophys. Res. Atmos., 121, 5159–5176. Quan, X., M.P. Hoerling, J. Perlwitz, H.F. Diaz, and T. Xu (2014), How Fast Are the Tropics Expanding?, J. Climate, 27, 1999–2013.

Schmidt, D. F., & Grise, K. M. (2017), The response of local precipitation and sea level pressure to Hadley cell expansion, Geophys. Res. Lett., 44, 10,573–10,582.

Solomon, A., L. M. Polvani, D. W. Waugh, and S. M. Davis (2016), Contrasting upper and lower atmospheric metrics of tropical expansion in the Southern Hemisphere, Geophys. Res. Lett., 43, 10,496–10,503.

---

## Author Response (AR1)

In the reply, the referee's comments are in *italics*, our response is in normal text, and quotes from the manuscript are in blue.

***Anonymous Referee #1***

***General comments***
*This manuscript is a valuable contribution to the geoengineering literature, as it provides a detailed assessment of the simulated tropical circulation response to uniform solar dimming in a suite of coupled climate models. The Hadley circulation does not return to preindustrial conditions in a climate with quadrupled carbon dioxide levels and reduced insolation (the G1 experiment). The authors attribute this result to changes in meridional temperature gradients rather than changes in static stability. The Walker circulation, by contrast, is largely restored to its preindustrial state in G1. The Introduction section effectively describes the many motivations for the study.*

*The novelty of this work lies in its assessment of the G1 experiment, as much analysis of the impact of elevated atmospheric carbon dioxide concentrations on tropical dynamics has been discussed previously (see references below). At present, the Introduction mentions a few studies on the latter subject (on page 3), but a more thorough review of the existing literature is warranted, both in the Introduction and the Discussion. This manuscript would be more effective if it were shortened so as to emphasize new knowledge; if the circulation changes in abrupt4xCO2 simulations differ from what is reported in the existing literature on the subject, this can be emphasized, but otherwise the geoengineering results should be brought to the forefront.*

**Reply:** Thanks for the suggested additional references. We modify the introduction to include many suggested by both referees:

Climate model simulations with increased greenhouse gas forcing also indicate a poleward expansion of the Hadley circulation, (Hu et al., 2013; Ma and Xie, 2013; Kang and Lu, 2012; Davis et al., 2016). Vallis et al. (2015) analysed the response of 40 CMIP5 climate models finding that there was only modest model agreement on changes. Robust results were slight expansion and weakening of the winter cell Hadley circulation in the Northern hemisphere. It is unclear how closely the model simulations match reality. Choi et al. (2014) and Quan et al. (2014) both suggest that reanalysis trends for the Hadley cell edges may be overstated, especially compared to independent observations., and model trends are in reasonable agreement with the reanalysis trends (Davis and Birner, 2017; Garfinkel et al., 2015), but choice of metric also matters (Solomon et al., 2016) when discussing trends.

Many authors have considered the impact of greenhouse gas forcing on the Hadley circulation, particular in respect of changes in the width of the tropical belt (e.g., (Frierson et al., 2007; Grise and Polvani, 2016; Johanson and Fu, 2009; Lu et al., 2007; Seidel et al., 2008), but far fewer have discussed changes in Hadley intensity (Seo et al., 2014; He and Soden, 2015). The importance of tropical belt widening is of course due to its impact on the hydrological system, especially the locations of the deserts (Lau and Kim, 2015; Seager et al., 2010), which are a critically important for the habitability of several well-populated areas.

We try to shorten the parts related to abrupt4×CO₂, but as the manuscript needs to be self-contained, we do need to discuss the greenhouse gas forced changes as well as the geoengineering results to some extent.

*I have questions pertaining to the methodological choices described in Section 2.3. Why is the Hadley cell intensity based on such a broad latitudinal extent (to 40° S or N)? This extends beyond the tropics and includes the Ferrel cell. Additionally, the Hadley cell migration is not symmetric in the two seasons (the July-September cell extends further into the summer hemisphere than the January-March cell), so why is the Hadley cell intensity metric hemispherically symmetric?*

**Reply:** We show how the Hadley intensity varies with 2 other choices of cell width in Table R1 below, using 38°-15° or 35°-15° extent in both hemispheres for each model.

There is a consistent difference of 7% for 38°-15° and 14% 35°-15° in all 3 experiments. This shows that while there are differences in the actual intensities computed depending on the latitudinal width of the cells, the results do not show differences between models or experiments. There is just a systematic offset in the intensity calculated, not a change of type of response calculated, or big across-model differences in behaviour, and the offsets are the same for the 3 experiments and both hemispheres. So even though the referee is correct, that we use none-standard definitions so that we capture all the variability in the Hadley cells in all the models and experiments, and also use symmetric cells, it seems that the changes in behavior we observe due to the experiments would not be affected.

**Table R1: The differences ($10^{10}$ kg s$^{-1}$) relative to the method used in the manuscript, defined as 40°-15° S and N) for the Hadley cells. The number in brackets is the percentage difference percent relative to our method**

| model | Southern hemisphere Hadley intensity in JAS defined by 38°S-15°N | | | Southern hemisphere Hadley intensity in JAS defined 35°S-15°N | | |
|---|---|---|---|---|---|---|
| | piControl | G1 | 4xCO2 | piControl | G1 | 4xCO2 |
| BNU-ESM | 7.4(7%) | 7.3(6.8%) | 7.7(6.9%) | 7.9(14.4%) | 7.8(14%) | 8(14%) |
| CanESM2 | 7(7%) | 6.8(7%) | 7.2(6.9%) | 7.5(14.3%) | 7.2(14.4%) | 7.7(14%) |
| CCSM4 | 7.7(6.5%) | 7.6(6.4%) | 7.9(6.3%) | 8.2(13.4%) | 8(13.3%) | 8(13%) |
| GISS-E2-R | 6.8(6.4%) | 6.4(6.4%) | 7(6.8%) | 7.2(13.5%) | 6.8(13.4%) | 7.5(14%) |
| HadGEM2-ES | 7.4(6.7%) | 7.4(6.6%) | 7.6(6.8%) | 8(14.4%) | 8(14%) | 8(14%) |
| IPSL-CM5A-LR | 6.4(7%) | 6.4(7%) | 6.6(7.5%) | 7(15%) | 6.8(14.7%) | 7(15.5%) |
| MIROC-ESM | 6.8(7.4%) | 6.4(7.5%) | 7.2(7.4%) | 7.4(15.5%) | 7(15.7%) | 7.7(15%) |
| NorESM1-M | 7(6%) | 6.8(6%) | 7(6%) | 7.5(12.8%) | 7.3(13%) | 7.7(12.8%) |
| Ensemble | 7(6.8%) | 7(6.7%) | 7.3(6.8) | 7.6(14%) | 7.4(14%) | 7.8(14%) |

| model | Northern hemisphere Hadley intensity in JFM defined 15°S-38°N | | | Northern hemisphere Hadley intensity in JFM defined 15°S-35°N | | |
|---|---|---|---|---|---|---|
| | piControl | G1 | 4xCO2 | piControl | G1 | 4xCO2 |
| BNU-ESM | 6.8(6%) | 6.4(6%) | 6.7(6.2%) | 7.2(13%) | 6.8(12.7%) | 7(12.6%) |
| CanESM2 | 6.3(6.3%) | 6(6.4%) | 6.2(6.3%) | 6.7(13%) | 6.5(13%) | 6.6(12.8%) |

| | | | | | | |
|---|---|---|---|---|---|---|
| CCSM4 | 7.0(6%) | 7(6%) | 7(6%) | 7.4(12.8%) | 7.3(12.8%) | 7.6(12.7%) |
| GISS-E2-R | 4.7(7%) | 4.6(7.4%) | 4.7(7.4%) | 5(15.4%) | 5(15.8%) | 5(16%) |
| HadGEM2-ES | 6.5(6.7%) | 6.3(6.7%) | 6.3(6.8%) | 7(14%) | 6.8(14%) | 6.7(14%) |
| IPSL-CM5A-LR | 5.3(7.6%) | 5.2(7.4%) | 5.2(8.2%) | 5.7(16%) | 5.6(15.7%) | 5.7(17%) |
| MIROC-ESM | 4.6(8.6%) | 4.5(8.7%) | 4.4(9%) | 5(18%) | 5(18.4%) | 4.8(18.4%) |
| NorESM1-M | 6.4(5.5%) | 6(5.6%) | 6.5(5.7%) | 6.8(11.4%) | 6.5(11.7%) | 6.8(11.8%) |
| Ensemble | 6(6.6%) | 6(6.7%) | 6(6.8%) | 6.4(13.8%) | 6.2(14%) | 6.3(14%) |

We add "We experimented with using narrower definitions of the Hadley cell (38°-15° or 35°-15°) in the 3 experiments, finding almost the same systematic offsets in intensities across the models and experiments. This is also true for each hemisphere separately. Departures in model ensemble mean intensity across the three experiments for both hemispheres from an outer latitude of 40° range from 6.6-7% and 13.8-14% with outer latitudes of 38°and 35° respectively. So using the wide latitude bands we chose captures all the variability in the Hadley cells in all the models and experiments without introducing biases due to experiments or hemispheres." in line 230.

*Finally, throughout the paper it would be helpful to explicitly distinguish between robust results and results for which the ensemble mean is dominated by inter-model cancellation. For example, it is important to elaborate on this last sentence of Section 3.1 (line 253). Does the substantial inter-model spread undermine subsequent interpretation of the ensemble mean change?*

**Reply:** We use standard statistical significance tests to quantify robustness of result throughout, e.g. in Table 2 where we distinguish between ensemble mean changes that are significant at 95% confident level in abrupt4×$CO_2$, but not G1. The other method commonly used to assess model agreement is if some super-majority, e.g.75% of models agree on the sign of an anomaly. Using that measure implies that in Table 2 the change under abrupt4×$CO_2$ is not robust. We add the model agreement wherever relevant to the results we present, for example we change:

"However there is some scatter between models (Table 2)." To:

There is significant change in the ensemble mean position and strength under abrupt4×$CO_2$, but not G1 in Table 2. However, only 5 out of 8 models agree on the sign of the changes, so the inter-model differences are rather large in this case.

*Specific comments*
*The ENSO-related results could be included in the abstract.*
*Throughout the paper, starting in the abstract, the Hadley circulation changes are discussed in terms of "seasonal maximum Northern and Southern cells." It would be clearer to discuss these together as the "solstitial Hadley cells," or as "the JAS and JFM cross-equatorial cells."*

**Reply:** Unfortunately the abstract length guide in TC is 100-200 words. Presently we

are at 250. We think that the changes in ENSO are of less importance than the other aspects we discuss in the abstract.

With respect we disagree that using the term *solstitial* is an improvement or disambiguation. Typically *solstitial* refers to the summer solstice (e.g. Websters disctionary), whereas in this case it would refer to the winter solstice. The vast majority of the cell we refer to "Northern" is indeed in the Northern hemisphere, and we think using the terms "JAS Southern" and "JFM Northern" cells is the least ambiguous choice.

*(lines 79-80) It is not entirely clear what is meant by this phrase: "and a similar response of oceans versus land."*
**Reply:** We change "There is also a relative undercooling of the polar regions and overcooling of the tropics and a similar response of oceans versus land with globally uniform SRM." To
The general pattern of temperature change under all abrupt4×CO$_2$ includes accentuated Arctic warming, and least warming in the tropics. G1 largely reverses these changes, but leaves some residual warming in the polar regions and under-cools the tropics relative to piControl. Geoengineering also reduces temperatures over land more than over oceans relative to abrupt4×CO$_2$, and hence reduces the temperature difference between land and oceans by about 1°C.

*(line 84) Held and Soden (2006) are better known for explaining this P-E scaling, which is derived from the Clausius-Clapeyron relation and only valid over ocean (reference below).*
**Reply:** Thanks, we add the reference to Held and Soden, we did not mean to claim priority for Tilmes, it was a convenient reference that we used elsewhere

*(lines 108-110) This sentence is unclear: "The signal to noise ratio [. . .]."*
**Reply:** We change "The signal to noise ratio in the G4 experiment is relatively low with a background of only the modest RCP4.5 greenhouse gas forcing scenario." To
The geoengineering and greenhouse gas forcing in the G4 experiment are both relatively low compared with the G1 experiment, since under G4 the greenhouse gas scenario is the modest RCP4.5, which means that natural climate variability in the 50 year long period of geoengineering period may obscure features.

*(line 198-200) Are all 50 years used only for those measures that do not rely on sea surface temperature?*
**Reply:** No. we use 50 years for zonal and meridional stream function and sea surface temperature correlation with STRF of the Walker circulation in Fig. 11. We note that There is no difference in model behavior between the G1 and abrupt4×CO$_2$ anomalies and △SST explains 83% of the overall variance. Despite a temperature transient of at a decade or so (e.g. Kravitz et al., 2013) in the abrupt4×CO2 simulation and the lack of any transient in STRF (Fig. S1), the relationship with ΔSST is nearly as good as for piControl. This is because the STRF depends on the SST at that time, which we correlate.

We use 30 years for air temperature and land temperature in Figs. 12 & 13.

*(line 213) Does the phrase "whole streamfunction" mean for all longitudes? Say that explicitly.*
**Reply:** Yes, and corrected

*(line 237) Does mean state refer to annual mean state?*
**Reply:** Yes, and corrected

*Labeling multi-panel figures would facilitate the discussion of results in the text.*
**Reply:** OK. We add letters to distinguish the separate panels.

*(line 308) What constitutes a "good relationship?"*
**Reply:** We clarify this as : Fig. 7 (B) shows that the modelled motion of the ITCZ explains 73% of the variance in intensity of the JAS Southern cell peak intensity, which is significant at the 95% level. Thus the larger the model reduction in intensity the more the boundary of the ITCZ moves equatorward.

*(line 312-314) Note that this was reported by Smyth et al. (2017).*
**Reply:** Noted: The combined seasonal effect of both cell changes is a reduced migration of the upwelling branches of the circulation cells across the equator, as was also noted by Smyth et al. (2017).

*(lines 321-344) This section can be made more concise by focusing on similarities or differences from previous studies on the subject (cf. major comment above).*
**Reply:** we modify this section as suggested:

The situation under abrupt4×$CO_2$ is more complex (Fig. 6 (E) and (F)). The expansion of the tropics has been noted both in greenhouse gas simulations and observationally (Davis et al., 2016; Hu et al., 2011), along with the larger southern expansion. The extratropical changes in the Ferrel circulation are also more pronounced in the Southern hemisphere.

Reduction in strength of the Northern hemisphere winter cell was also a robust result of climate models under RCP8.5, while, the Southern cell exhibited almost no change (Vallis et al., 2015). Our results in Fig. 8 show that the multi-model ensemble mean reduction Hadley intensity under G1 of -18×$10^8$ kg s$^{-1}$ and of -7×$10^8$ kg s$^{-1}$ for abrupt4×$CO_2$. The JAS Southern Hadley intensity exhibits a fall of -16×$10^8$ kg s$^{-1}$ under G1 but an increase of 23×$10^8$ kg s$^{-1}$ under abrupt4×$CO_2$. At least 6 out of 8 models agree on these sign of changes in both hemispheres and scenarios. Thus the Southern hemisphere results differ for abrupt4×$CO_2$ from those presented in Vallis et al. (2015). The anomalies for most models are significant, and the ensemble means are 8 standard errors from zero and thus very highly significant.

We move our definition of the anomalies to the fig 8 caption:

Figure 8. Anomalies ($10^{10}$ kg s$^{-1}$) relative to piControl amongst models in Hadley circulation for the Southern cell in JAS (left panel), defined as the magnitude of the

mean meridional stream-function between 15°N and 40°S, and (right panel) the Northern cell in JFM, defined as the magnitude of the mean meridional stream-function between 15°S and 40°N. The dot size for the models is about 1 standard error of the model mean.

*(lines 339-340) Cite other studies which have noted that solar dimming results in an overcompensation of tropical circulation changes induced by global warming.*
**Reply:** we don't see how this question arises from the text at line 339-340 nor anywhere else.

*You might consider discussing all of the analysis of the Walker and Hadley circulation responses together, i.e. moving current Section 6.1 to follow current Section 3.2.*
**Reply:** We want to discuss the mechanism of Walker and Hadley circulation by analyzing the relationship between them and temperature. Section 6 is where we do this as that is focused on mechanisms rather than presenting results as in section 3 and 4.

*(lines 399-403) This section is confusing. "Monthly temperatures" in which region?*
**Reply:** This sentence is not needed, as we discuss only annual means in the rest of the paper and so we delete it**.**

*(lines 433-436) Why do you choose to analyze temperatures over Tibet if the general land-ocean temperature contrast is of interest? This choice should be justified, or the analysis modified. Are you considering surface temperatures? The methodology is not described in enough detail for the results to be reproduced. Additionally, there is an extensive body of literature connecting inter-hemispheric temperatures and the Hadley circulation/Intertropical Convergence Zone, such as Broccoli et al. (2006), which can be referred to here.*
**Reply:** There is difference in surface temperature response of oceans and land under geoengineering. Under pure greenhouse gas forcing very small land-ocean temperature differences are forecast in the tropics, so it is interesting to explore any change in the circulation behavior that may result. There is also well-documented influence of Tibet-Indian Ocean temperature difference affecting Asian monsoon intensity. Hence it seemed worthwhile to explore this in the fig S9. We do think it is worthwhile to show this, though we don't discuss the result in detail in the paper. We modify the text:
Both SRM and greenhouse gas forcing modifies the land-ocean temperature difference relative to piControl and so conceivably affects Hadley circulation, for example by changing the hemispheric temperature and the position of the ITCZ (Broccoli et al., 2006). Under abrupt4×CO$_2$ land-ocean temperature differences in the tropics (between 30° N and 30°S) are reduced to essentially zero, while under G1 differences in the tropics are 1.2°C which is not significantly different from the piControl difference of 1.4°C. Since the largest continental land masses are in the Northern hemisphere, we would expect any differences in Hadley circulation induced by land-ocean contrasts in the Northern hemisphere to be visible in the Southern Hadley cell. We explored the impact of land-ocean temperature differences by considering differences in the surface

temperatures over Tibet and the whole tropical ocean temperature (Fig. S9). Results were similar as for Fig. 13, with significant correlations for G1 in the Southern Hadley cell.

*(lines 445-448) Specify the location and season on which this is based. Are these near-surface potential temperature gradients?*

**Reply:** As we use the Seo et at. (2014) method, the potential temperature gradients are defined here as the average between 1000 and 400 hPa. And we use JFM for Northern hemisphere and JAS for Southern hemisphere. We specify this in the Table 3 notes:

& Function 1 is $\frac{5}{2}\frac{\delta H}{H} + \frac{5}{2}\frac{\delta \Delta_H}{\Delta_H} - \frac{\delta \Delta_V}{\Delta_V}$ and is based the model of Held and Hou (1980).

**Function 2 is $\frac{9}{4}\frac{\delta H}{H} + 2\frac{\delta \Delta_H}{\Delta_H} - \frac{3}{4}\frac{\delta \Delta_V}{\Delta_V}$ which is derived from the model by Held (2000).**

$\Delta_H$ is meridional temperature gradient defined as $\frac{\theta_{eq} - \theta_{higher\,lat}}{\theta_0}$ which is the tropospheric mean meridional potential temperature gradient with $\theta_0$ denoting the hemispheric troposphere mean potential temperature and $\theta_{eq}$ calculated between 10°N and 10°S. We follow Seo et al. (2014) in taking $\theta_{higher\,lat}$ as the average potential temperature between 10°-50°N for the Northern hemisphere winter and 10°-30°S for the Southern hemisphere. Potential temperature gradients are defined here as the average between 1000 and 400 hPa. $\Delta_V = \frac{\theta_{300} - \theta_{925}}{\theta_0}$ is the dry static stability of the tropical troposphere. $H$ is the subtropical tropopause height estimated as the level where the lapse rate decreases to 2°C km[-1].

! The Hadley intensity $\psi_m$ is described in section 2.3 and we use JFM in the Northern hemisphere and JAS in the Southern hemisphere.

*(lines 468-470) The sentence "However only the eastern and western [. . .]" is confusing.*

**Reply:** Rewrite it as "However the eastern and western boundaries of the Walker circulation shift westward during El Niño in G1 relative to piControl."

*The analysis of the Walker circulation should be better framed. For example, He and Soden (2015) explain that weakening of the Walker circulation due to carbon dioxide forcing is mostly driven by the change in mean sea surface temperatures (SST).*

**Reply:** Thank you, we now include this as a frame to the discussion: He and Soden (2015) conclude from experiments designed to elucidate the role of various forcings on tropical circulation that weakening of the Walker circulation under greenhouse gas forcing is primarily due to mean SST warming. They also note that increased land-sea temperature contrast results in strengthening of the circulation, and also that while the pattern of greenhouse gas warming is close to an El Niño, there are sufficient differences to produce quite different responses in the Walker circulation. We may

therefore expect that changes under G1 compared with pure greenhouse gas forcing would manifest themselves given the changes in both the direct and indirect $CO_2$ forcings. What we observe though is that changes in the Walker circulation are modest, and examination of the dependence on intensity as a function of zonal Pacific Ocean temperature differences (Fig. 11) show no differences between the greenhouse gas and G1 forcings. Similarly we find no change in the intensity with land-ocean temperature gradients.

*(line 500) The second half of this sentence is unclear.*
**Reply:** we rewrite this as 2 sentences:
Furthermore the intensity of the Hadley circulation is expected to decrease as it expands and also in response to an accelerated hydrological cycle. An enhanced hydrological cycle is expected under greenhouse gas forcing, but not solar geoengineering which leads to net drying (Kravitz et al., 2013).

*(lines 514-516) Rephrase this sentence for clarity: "But we observe [. . .] G1 forcing."*
**Reply:** We change it to: Nor it is consistent with increases simulated in the Southern Hadley cell intensity and simultaneous decreases in the Northern one relative to piControl, although both are stronger than under the G1 forcing.

*(line 545) Does "ocean heating" refer to warming sea surface temperatures or ocean heat uptake?*
**Reply:** Yes, we rephrase this for clarity: intuitively mean reduced heating, sea surface temperatures and moisture flux

*(line 548) Define "Rx5day extreme"*
**Reply:** We were inaccurate in the phrase and change it as : shows that the annual wettest consecutive five days are drier

*Figure 10: Is this based on annual data from El Nino years, or data from a particular season? In a few places in the paper it is not immediately clear what averaging periods and spatial domains are used for calculations.*
**Reply:** The method is defined in section 2 lines 159-165. "Composite analysis is applied for the study on the influence of ENSO. We follow Bayr et al. (2014) and use detrended and normalized Nino3.4 index (monthly averaged sea surface temperature anomaly in the region bounded by 5° N - 5° S, from 170° W - 120° W) as a criteria to select ENSO event. An index > 1 represents an El Niño event and < -1 a La Niña one (Bayr et al., 2014). We concatenate variables in all El Niño and La Niña events for each individual model to get El Niño and La Niña data sets and then calculate ensemble results."

*Technical comments*
*(Line 10) capitalize "Earth"*

**Reply:** done.

*(Line 25) "good correlations" should be rephrased more precisely*
**Reply:** Change it as "There are significant relationships between Northern cell intensity and land temperatures"

*(Line 55) typically capitalized "Northern Hemisphere"*
**Reply:** done.

*(line 29) "response to" should say "responses of"*
**Reply:** done.

*(Line 63) "compliment" should be "complement"*
**Reply:** done.

*(line 157) not a sentence*
**Reply:** Changed to We used monthly-mean model output data.

*(line 349-351) This is not a sentence.*
**Reply:** Changed to Hadley circulation shrinks and strengthens during El Niño events, while expanding and weakening during La Niña.

*(line 542) "While under [. . .]" is not a sentence.*
**Reply:** Changed to: There are clear changes of Hadley cells under the latitudinal varying forcing of G1.

**Additional modifications:** We also revise the following sections.
(Line 59) Rewrite the sentence.
Model results suggest a significant eastward movement with weakening intensity under greenhouse gas forcing (Bayr et al., 2014), and He and Soden (2015) propose that the sea surface temperature warming plays a crucial role in both the eastward shift and the weakening of Walker circulation. They also note that this weakening may be reversed by rapid land warming.

(Line 207) Correct the writing mistake (5°S - 5°N, 160°W - 80°E) to (5°S - 5°N, 160°W - 80°W)

(Line 450) We change "respectively in south and north cell" to "respectively in Southern and Northern cells"

(Line 475) Delete "Davis et al., (2016) note an expansion in the Hadley cells in

proportion to the temperature rises in the models under both G1 and abrupt4×CO2."

(Line 543) Rewrite it as "The reduction in incoming shortwave radiation in G1 would intuitively mean reduced heating, sea surface temperatures and moisture flux in the ITCZ"

(Line 545) Delete "Reduced ocean heating would then tend to mean a smaller amplitude of seasonal movement of the ITCZ"

(Line 269-273) We redraw Fig.1 to show the difference of ERA-piControl rather than piControl-ERA and rewrite the relevant text "Fig. 1 (D) shows that the ERA-Interim circulation has an eastward displacement and the intensity measured by STRF is overestimated by 26% relative to ensemble piControl. There is a similar structure to the stream function differences between NCEP2 reanalysis and piControl, and the STRF is only overestimated by 3% relative to ensemble piControl."

(Line 316-317) We redraw the Fig.4 show the difference of ERA-piControl rather than piControl-ERA and rewrite the relevant text "The intensity anomalies relative to piControl from both the reanalysis data sets are less than 21% (Fig. 4)."

In the reply, the referee's comments are in *italics*, our response is in normal text, and quotes from the manuscript are in blue.

*Anonymous Referee #2*

*General comments*

*While the introduction has sufficient breadth, there are areas where it lacks a discussion of more recent work. For example, regarding the statement ". . .climate model simulations. . .indicate a poleward expansion of the Hadley circulation, though weaker than that observed", there numerous studies suggesting that this may not be the case. Choi et al. 2014 and Quan et al. 2014 both suggest that reanalysis trends in the HC edges may be overstated, especially compared to independent observations. And it appears that the model trends may not be so different from the reanalysis trends (Garfinkel et al. 2015, Davis and Birner 2017). The choice of metric also matters (Solomon et al. 2016). My understand is that it's actually unclear whether models, reanalyses, and observations disagree on the response of the Hadley circulation; but that itself is a valid motivation. The authors may also consider connecting their work to studies like Schmidt and Grise (2016), who have investigated some of the longitudinal characteristics of Hadley cell variability. Full references can be found at the end.*

**Reply:** Thanks for these suggestions. We modify the introduction to include many suggestions by both referees:

Climate model simulations with increased greenhouse gas forcing also indicate a poleward expansion of the Hadley circulation, (Hu et al., 2013; Ma and Xie, 2013; Kang and Lu, 2012; Davis et al., 2016). Vallis et al. (2015) analysed the response of 40 CMIP5 climate models finding that there was only modest model agreement on changes. Robust results were slight expansion and weakening of the winter cell Hadley circulation in the Northern hemisphere. It is unclear how closely the model simulations match reality. Choi et al. (2014) and Quan et al. (2014) both suggest that reanalysis trends for the Hadley cell edges may be overstated, especially compared to independent observations, and model trends are in reasonable agreement with the reanalysis trends (Davis and Birner, 2017; Garfinkel et al., 2015), but choice of metric also matters (Solomon et al., 2016) when discussing trends.

Many authors have considered the impact of greenhouse gas forcing on the Hadley circulation, particular in respect of changes in the width of the tropical belt (e.g., (Frierson et al., 2007; Grise and Polvani, 2016; Johanson and Fu, 2009; Lu et al., 2007; Seidel et al., 2008), but far fewer have discussed changes in Hadley intensity (Seo et al., 2014; He and Soden, 2015). The importance of tropical belt widening is of course due to its impact on the hydrological system, especially the locations of the deserts ( Lau and Kim, 2015; Seager et al., 2010), which are a critically important for the habitability of several well-populated areas.

*I think the GISS-E2-R model should be excluded from the composite figures, while the composites with GISS should be shown in the supplementary information (opposite to what is currently done). The authors have made a good choice to show composites with*

*and without GISS, as its behavior deviates so much from the other models, but I think swapping these figures would better support their conclusions and interpretations while still maintaining full disclosure.*

**Reply:** We thought long about doing this. Although GISS-E2-R model anomalies in G1 and abrupt4xCO2 are different from other models they more or less plausibly describe Walker and Hadley circulation structure. We felt that it would be better to include the GISS model results in the ensembles we show in the main text simply because of the virtue of including more models. The GISS model does not unduly affect the ensemble means, and while in some cases it does affect the model spread, this is perhaps more representative of actual uncertainties when the models attempt to simulate the large forcings in G1 and abrupt4$\times$CO$_2$.

*It would be helpful if the authors were more explicit and definitive throughout the manuscript. For example, on lines 97-100, it would be helpful to readers if the sign of the changes were stated, e.g., ". . .report that decreases in Hadley cell intensity drive the reduction in tropical precipitation under solar geoengineering. . .". Or, on lines 144- 146, state whether the abrupt4xCO$_2$ experiment is close to RCP8.5 in terms of CO2 ppm or in terms of radiative forcing. What follows is a list of some but not all of these instances: -Line 74 -Line 80 -Lines 103-104 -Lines 223-224 -Lines 251-252 -Lines 293-294 -Lines 337-340 (are intensity changes signed, or in terms of absolute value?) -Lines 480-482*

**Reply:** Thanks, we have made the changes suggested:

97-100: report that decreases in Hadley cell intensity drive the reduction in tropical precipitation under solar geoengineering

144-146 an atmospheric CO$_2$ concentration of nearly 1140 ppm, close to concentrations under "business as usual" scenarios such as RCP8.5

74 produce net drying due to the decreasing in vertical temperature gradient

80 The general pattern of temperature change under all abrupt4$\times$CO$_2$ includes accentuated Arctic warming, and least warming in the tropics. G1 largely reverses these changes, but leaves some residual warming in the polar regions and under-cools the tropics relative to piControl. Geoengineering also reduces temperatures over land more than over oceans relative to abrupt4$\times$CO$_2$, and hence reduces the temperature difference between land and oceans by about 1°C.

103-4 This tropical circulation pattern is intimately related to changes in the Walker circulation by their dependences on the Pacific Ocean zonal sea surface temperature gradient

223-4 The boundary at the edge of the tropics is also known to move latitudinally but the circulation cell rapidly becomes weaker beyond the zero crossing of the rotation sense.

251-2 This sentence is not relevant so we delete it.

293-4 The Southern cell shows a complex anomaly structure with positive anomaly between 45°S-65°S also in the Ferrel cell circulation that borders it at higher southern latitudes.

337-40 We have defined the Hadley circulation intensity in section 2.3 as we write in

line 334-336
480-482 Beyond the Hadley cells there are modest, but statistically significant changes, particularly in the Southern hemisphere Ferrel circulations with poleward movement.

*I have difficulty discerning information from the anomaly contour figures, like Fig. 2. Standard practice is to show control values overlaid as contours on the shading, so that shifts/expansions/contractions can be more easily discerned. This is especially critical for the Walker circulation - its mean structure and response have substantial spatial variability*
**Reply:** Figure2, 5, 6 have been redrawn with piControl contours overlaid anomalies shading.

*The questions posed at the end of the introduction are a great way to orient the reader. It may be worth specifically restating these questions in the discussion as a way of summarizing the results.*
**Reply:** To answer the questions we posed in the introduction we reorganized our paper and add a summary section:

Our main purpose in this study has been to answer the following questions: Does the G1 scenario counteract position and intensity variations in the Walker and Hadley circulations caused by the greenhouse gas long wave forcing under abrupt4×CO$_2$? How does the tropical atmospheric circulation, including the Walker and Hadley circulations, respond to warm and cold phases of the El Niño Southern Oscillation (ENSO) in G1 and abrupt4×CO$_2$?

The Walker circulation in G1 displays insignificant increases in intensity and no shift in its western edge in the Pacific Ocean relative to piControl and hence does counteract the changes from greenhouse gas forcing. There is a potentially important change in position of the Walker circulation associated with the West African rainforest and East African grassland zones under G1, with potential for the encroachment of a drier climate into the Congo basin. In contrast, the Hadley circulation shows larger changes under G1 that are not simple reversals of those induced by greenhouse gas forcing on piControl climate. There is an asymmetric response between the hemispheres under both greenhouse gas and solar dimming that are correlated with direct forcings rather than adjustment of sea surface temperatures, and correlated with changes in meridional and land-ocean temperature gradients. These differences in response of the Hadley and Walker circulations are consistent with the zonally invariant forcing of both solar dimming and greenhouse gases and the meridionally varying solar dimming.

A clear Walker circulation westward movement during El Niño and an eastward movement during La Niña are shown nearly everywhere along the equator in abrupt4×CO$_2$. However the eastern and western boundaries of the Walker circulation shift westward during El Niño in G1 relative to piControl. The range and amplitudes of significant changes are smaller in G1 than in abrupt4×CO$_2$. The same is true in general for the Hadley cell. Under abrupt4×CO$_2$ the Northern Hadley cell significantly decreases in intensity under both la Niña and El Niño conditions while under G1 the

decreases are smaller and limited to each cell's poleward boundaries.

Both models and the limited observational data available on the Hadley circulation indicate that it is not zonally symmetric: there are intense regions of circulation at the eastern sides of the oceanic basins (Karnauskas and Ummenhofer, 2014), while elsewhere circulation is reversed, and much of the natural variability of the circulation is related to ENSO (Amaya et al., 2017). This and the opposite correlations with surface temperatures in the Pacific and SPCZ with STRF under G1 (Fig. 12) suggests an interplay between Hadley and Walker circulations that could repay further consideration of model data at seasonal scales. The importance of the tropical ocean basins as genesis regions for intense storms also suggests that changed radiative forcing there under geoengineering could cause important differences in seasonal precipitation extremes, that may be hidden in monthly or annual datasets.

*The figure production quality is high, but the image quality is low. Per ACP guidelines, PDF or EPS is preferred so that the figures are crisp when zoomed in (save with vector graphics enabled; in MATLAB, it's the "painters" renderer, not sure how this works in IDL or other languages). Otherwise, I think the DPI needs to be increased for the figures.*
**Reply:** The figures are standard pdfs as made, and passed the quality control for ACP. The final version figures would be higher resolution we expect. But the figures zoom quite well to 300% or more in our viewer.

*Specific comments*
*Line 65: Define "SRM" here rather than on line 73.*
**Reply:** done

*Line 99: What are the seasonal changes?*
**Reply:** we rephrase: and that seasonal changes mean that the ITCZ has smaller amplitude northward shifts compared with no geoengineering.

*Line 135: ENSO was already previously defined.*
**Reply:** Deleted here

*Lines 191-194: What is the order of the variability in the first 3 years - 1 sigma, 2 sigma? This may help convince readers it's nothing to worry about.*
**Reply:** rephrased as: that have significantly (p<0.05) higher STRF in the first 10 years of the abrupt4×CO$_2$ simulation than in following decades. This is not due to a transient affecting the first few years, but rather to higher values around 3 years into the simulation, but this is not unusual for each model's multiannual and decadal variability.

*Lines 227-233: I am somewhat unclear on the metrics for Hadley cell intensity. It seems like the authors are using the average of the 900-100 hPa stream function; are they*

*using a point maximum or an area average?*

**Reply:** We rephrase for clarity: Thus we define the Hadley circulation intensity for the Southern cell as the average meridional stream-function between 900-100hPa over the area between 40°S and 15°N in July, August and September (JAS), and the Northern cell as the absolute value of mean meridional stream-function between 15°S and 40°N in January, February and March (JFM).

*Line 239, 283-285: "Intuitively": "effectively" or "naturally" might be more appropriate?*

**Reply:** We prefer the version as it on line 239. We rephrase line 283 as This can naturally describe

*Line 243: Mentioning a specific number is good, as is describing how it is derived.*

**Reply:** This seems clear from the text the intensity measured by STRF is underestimated by 3% relative to ERA-Interim. This number is the relative change between piControl and ERA

$$\frac{STRF_{piCotrol} - STRF_{ERA-Interim}}{STRF_{ERA-Interim}}$$

*Lines 263-266: The scatter here is very large among the models, even neglecting GISS, which I think is worth noting.*

**Reply:** yes this is true, a similar point was raised by ref#1. Another method commonly used to assess model agreement is if some super-majority, e.g.75% of models agree on the sign of an anomaly. Using that measure implies that the change under abrupt4×CO₂ is not robust. We add the model agreement where relevant to the results we present, for example we change:

"However there is some scatter between models (Table 2)." To:

There is significant change in the ensemble mean position and strength under abrupt4×CO₂, but not G1 in Table 2. However, only 5 out of 8 models agree on the sign of the changes, so the inter-model differences are rather large in this case.

*Lines 289-290: It might be more effective to say that there is enhanced overturning aloft and weakened overturning at lower levels; as written, it almost sounds like the anomalies don't conserve mass (reduced equatorward + enhanced poleward).*

**Reply:** Agree. We change it as "Circulation anomalies under abrupt4×CO₂ (Fig. 5 (B)), show enhanced overturning aloft and weakened overturning at lower levels."

*Line 309: How is the ITCZ metric defined?*

**Reply:** This defined in the caption to fig. 7, cited on line 308: The ITCZ position is defined from the centroid of precipitation (Smyth et al., 2017).

*Lines 321-325: I think this was essentially said previously on lines 288-294.*

**Reply:** yes, we delete the repeated section and rewrite according to suggestions of ref#1.

The situation under abrupt4×CO₂ is more complex (Fig. 6 (E) and (F)). The

expansion of the tropics has been noted both in greenhouse gas simulations and observationally (Davis et al., 2016; Hu et al., 2011), along with the larger southern expansion. The extratropical changes in the Ferrel circulation are also more pronounced in the Southern hemisphere.

Reduction in strength of the Northern hemisphere winter cell was also a robust result of climate models under RCP8.5, while, the Southern cell exhibited almost no change (Vallis et al., 2015). Our results in Fig. 8 show that the multi-model ensemble mean reveals a diminished Northern Hadley, intensity under G1 of $-18\times10^8$ kg s$^{-1}$ and of $-7\times10^8$ kg s-1 for abrupt4×CO$_2$. The Southern Hadley intensity in JAS exhibits a fall of $-16\times10^8$ kg s$^{-1}$ under G1 but an increase of $23\times10^8$ kg s$^{-1}$ under abrupt4×CO$_2$. Thus the Southern hemisphere results differ for abrupt4×CO$_2$ from those presented in Vallis et al. (2015).

*Line 333: Could write simply "more pronounced in the Southern hemisphere", as the "than. . ." is implied.*
**Reply:** Deleted, as shown above.

*Line 341: I think this is too colloquial, maybe state the p-value or % significance it reaches - stating something like "99.9% significant" is more convincing than "hugely"*
**Reply:** The exact significance level is very hard to define given the lack of knowledge of the probability distribution at the extreme tails. We change the text: The anomalies for most models are significant, and the ensemble means are 8 standard errors from zero and thus very highly significant.

*Lines 399-400: What is meant by "monthly temperature"?*
**Reply:** Monthly correlations between STRF and surface temperatures. These are not relevant to the rest of the manuscript and this sentence is deleted.

*Line 409: Which experiments? Citation?*
**Reply:** The citation is Van der Wiel et al., 2016, we were not clear how this was linked to the next sentence. We rephrase this part as: Experiments with an atmospheric circulation model (Van der Wiel et al., 2016) suggest that a key feature of the diagonal structure of the SPCZ is the zonal temperature gradient in the Pacific which allows warm moist air from the equator into the SPCZ region. This moisture then intensifies (diagonal) bands of convection carried by Rossby waves (Van der Wiel et al., 2016).

*Lines 413-415: Suggest stating model names, or at least specifying "two models" instead of "three. . .except BNU"*
**Reply:** Rewritten as "Two of the three models with positive correlation between STRF and SPCZ temperatures, CCSM4 and NorESM1-M, have increased STRF and ΔSST under G1"

*Lines 420-421: But isn't it the case that there are still some weak, positive correlations in regions like the SPCZ?*

**Reply:** Yes, there are some weak, positive correlations in some region under IPSL-CM5A-LR, MIROC-ESM and HadGEM2-ES. But here we are focusing the most obvious difference in these 3 model compared to others.

*Section 6.2/Table 3: I suggest mentioning the highest and lowest model value for each category, or really anything to help illustrate the model spread. With an N of 4, the average doesn't mean as much, but I think this section is still worth including. For the critical relationships, it may help to show them in scatter plots. I'm curious what output fields are needed that are lacking in most of the models?*

**Reply:** Yes we include the range from the 4 models in Table 3

| Scenario | G1-piControl | | abrupt4×CO$_2$-piControl | |
|---|---|---|---|---|
| | North | South | North | South |
| Temperature gradient | -2.6 (-3.5 — -1.1) | -1.2 (-1.7 — 0.1) | -4.4 (-6.1 — 0.7) | -4 (-6.1 — -0.3) |
| Static stability | -3.4 (-4.7 — -1.5) | -3.2 (-5.2 — -0.4) | 21 (18 — 26) | 23 (21 — 27) |
| Subtropical tropopause height | -0.1 (-2.1 — 1.8) | -0.5 (-1.4 — -0.1) | 0.87 (1.2 — 6) | 3 (-0.7 — 4) |
| Function 1 | -3.35 (-9.8 — 4.4) | -1.05 (-7.5 — 1.2) | -29.8 (-30 — -17) | -25.5 (-32. — -19) |
| Function 2 | -2.9 (-8.2 — 3.8) | -1.13 (-6.4 — 0.7) | -22.6 (-23 — -12) | -18.5 (-24 — -14) |
| Hadley intensity | -3.7 (-6.4 — -0.5) | -1.2 (-6 — 0.8) | -3.4 (-4.1 — -1) | 4.3 (2.4 — 4.8) |

*Lines 513-514: Why is this expected? Does it follow from the vertical expansion, or from the Held and Soden static stability/Clausius-Clapeyron scaling?*

**Reply:** The two scaling theories from Seo et al., (2014) listed in Table 3 indicate that the HC strength is proportional to the tropopause height and equator-to-higher-latitude potential temperature gradient. However, most climate models predict both increases in tropopause height and decreases in intensity under greenhouse gas forcing. Vallis et al (2015) summarize the argument as: "A general weakening of the tropical circulation might be expected from thermodynamic and energetic arguments involving water vapour concentration and precipitation (Boer, 1993; Held and Soden, 2006) and reviewed by Schneider et al. (2010). In brief, unless changes in relative humidity are very large, changes in the water vapour content of the atmosphere are mainly determined by changes in the saturation vapour pressure and hence by the Clausius–Clapeyron relation, and so increase by about 7% K−1. However, maintaining a surface energy balance constrains the changes in evaporation and precipitation to be closer to 3% K−1. Thus, the overall water vapour turnover rate will decrease as surface temperature increases, possibly leading to a weakening of the atmospheric circulation, and in particular the tropical circulation – at least to the degree that the circulation is controlled by such an effect. It is however by no means clear that the dynamics of the Hadley Cell is so controlled."

The situation is far from clear theoretically, and so we reflect this in modified text: We note that the robustly understood vertical expansion of the circulation as the tropopause rises under abrupt4×CO$_2$, has been associated with a decrease in the circulation

intensity (Seo et al., 2014; He and Soden, 2015) in climate models forced by greenhouse gases, and as expected from considerations of Clausius-Clapeyron scaling if relative humidity is relatively constant, as summarized by Vallis et al. (2015). This is not the case for the scaling functions from Seo et al., (2014; Table3), where tropopause height change is proportional to intensity change. Nor it is consistent with increases simulated in the Southern Hadley cell intensity and simultaneous decreases in the Northern one relative to piControl, although both are stronger than under the G1 forcing.

*Lines 520-524: Grise and Polvani (2016) would be a good reference for the dynamical response in abrupt4xCO2 outpacing the thermodynamic response. Doesn't this call into question the importance of static stability and meridional gradients in driving the changes in the circulation, if the circulation responds faster? Is it possible that the thermodynamic responses the authors examine might follow from some of the circulation changes, as these circulations transport heat?*

**Reply:** Thank you very much. This issue has indeed concerned us as well. We were hesitant to put it into our manuscript, but now we have made the statement following your suggestion.

Grise and Polvani (2016) explored how the dynamic response of the atmosphere, including metrics such as Hadley cell edge, varied with model climate sensitivity, that is the mean temperature rise associated with doubled $CO_2$. They found significant correlation across a suite of CMIP5 models running the abrupt4×$CO_2$ were largely confined to the Southern hemisphere, and also that the pole-to-equator surface temperature gradient accounted for significant parts of the dynamic variability that was not dependent on the mean temperature. However, we find that the response times of the Hadley circulations to changes in radiative forcing are very fast, as shown by the lack of transients in the simulated time series. Sea surface temperatures, especially under the strong abrupt4×$CO_2$ forcing takes at least a decade and parts of the system, such as the deeper ocean, would require even longer to reach equilibrium. Under abrupt4×$CO_2$ the global land-ocean temperature difference is reduced by about 1.3°C relative to piControl, while G1 reduces the contrast by only 0.3°C. The Northern hemisphere continents have faster response times than the oceans and so we would expect the Southern hemisphere to be much further from an equilibrium response than the Northern. This is also reflected in the lack of an equivalent to the "Arctic amplification" seen in the Northern hemisphere under both observed and simulated forcing by greenhouse gases. The lack of anomalous Southern polar warming is linked to the much cooler surface temperatures in the Antarctic mitigating against both temperature feedbacks and the ice-albedo feedback mechanism (Pithan and Mauritsen, 2014). The speed of response of the circulation changes calls into question the importance of static stability and meridional gradients in driving the changes in the circulation, since the circulation responds faster. Bony et al. (2013) attributed rapid changes in circulation in quadrupled $CO_2$ as due to direct $CO_2$ forcing. Fast response could also be a result of cloud feedback, land-ocean temperature differences and perhaps humidity, which are also important for poleward energy transport in G1 (Russotto and Ackerman, 2018; Russotto and Ackerman, in review ACP). Low cloud

fraction decrease under G1, warming the planet by reducing the reflection of solar shortwave radiation, but atmospheric humidity is reduced allowing heat to escape, and less energy is transported from tropics to poles.

*Line 548: What is "Rx5day"? I would rewrite this more generally, and avoid acronyms unless they will be useful later*
**Reply:** Yes agreed, rewritten as: shows that the annual wettest consecutive five days are drier

*Lines 553-554: "Intense" - subsidence?*
**Reply:** We rewrite this as: Both models and the limited observational data available on the Hadley circulation indicate that it is not zonally symmetric: there are intense regions of circulation at the eastern sides of the oceanic basins (Karnauskas and Ummenhofer, 2014), while elsewhere circulation is reversed, and much of the natural variability of the circulation is related to ENSO (Amaya et al., 2017).

*Figure 1 caption: Suggest rewriting so description doesn't only apply to the third subplot, i.e., "Walker circulation in the ERA-Interim reanalysis (top), . . . , model ensemble mean under piControl (third row), . . ."*
**Reply:** We will label the panels in the figure. And change the caption: Walker circulation in the ERA-Interim reanalysis (A), NCEP2 reanalysis (B), model ensemble mean under piControl (C) and difference between ERA-Interim and piControl (D). Color bar indicates the value of averaged zonal mass stream-function ($10^{10}$ kg s$^{-1}$). Warm color (positive values) indicate a clockwise rotation and cold color (negative values) indicate an anticlockwise rotation.

*Figures 7, 8, 11, 13: I think the authors have crafted the color scheme to avoid some of the major color-blindness combinations (i.e., no green/red), but a further helpful step is to not rely solely on color when trying to distinguish data points. I encourage the authors to use different symbols/shapes, in addition to their current color scheme, if they want to communicate values for each model individually rather than the behavior of the models as a whole*
**Reply:** In Fig. 7 this could be done, but the other figures would not work well. In Fig. 8 we want the shape (circle) to represent the size of the model standard error, these are the same for each model, having different shapes might confuse that point. Actually in figures 11 and 13 we want to distinguish between the G1 and the $4\times CO_2$ results by using different shapes, adding more shapes would results in much confusion and difficulty in recognizing 8 or more different shapes. Hence for simplicity and readability overall we prefer to keep the figures with simple shapes.

*Table 2: How is the percent change in position defined?*
**Reply:** First we define the position in section 2.2 as : we use the western edge of Walker circulation to represent it position. The western edge is defined by the zero value of the

vertically averaged $\psi_z$ between 400 – 600 hPa in the western Pacific 120°E – 180°E, (Ma and Zhou, 2016). Then we use the following function to calculate the percentage change relative to piControl. $x$ here refer to G1 or abrupt4xCO2 experiment.

$$\frac{Position_x - Position_{piControl}}{Position_{picontrol}}$$

We think that the table header: The number in the brackets represent percentage change relative to piControl explains this calculation.

*Table 3: The definitions should not be in the caption, but should instead be in the text.*
**Reply:** We prefer not to since the definitions are simply extracted from Seo et al. (2014), and would present a fairly large block of distracting definitions and equations that breaks the overall point of the section. Putting the definitions in the Table means they are included within the paper, but do not distract the reader, or give the impression that we have deduced them ourselves. The placing of the text, in either footnotes as we have now done, or the header as originally, we prefer to leave to the ACP editorial team.

**Additional modifications:** We also revise the following sections.
(Line 59) Rewrite this sentence.
Model results suggest a significant eastward movement with weakening intensity under greenhouse gas forcing (Bayr et al., 2014), and He and Soden (2015) propose that the sea surface temperature warming plays a crucial role in both the eastward shift and the weakening of Walker circulation. They also note that this weakening may be reversed by rapid land warming.

(Line 207) Correct the writing mistake (5°S - 5°N, 160°W - 80°E) to (5°S - 5°N, 160°W - 80°W)

(Line 450) We change "respectively in south and north cell" to "respectively in Southern and Northern cells"

(Line 475) Delete "Davis et al., (2016) note an expansion in the Hadley cells in proportion to the temperature rises in the models under both G1 and abrupt4×$CO_2$."

(Line 543) Rewrite it as "The reduction in incoming shortwave radiation in G1 would intuitively mean reduced heating, sea surface temperatures and moisture flux in the ITCZ"

(Line 545) Delete "Reduced ocean heating would then tend to mean a smaller amplitude of seasonal movement of the ITCZ"

(Line 269-273) We redraw Fig.1 to show the difference of ERA-piControl rather than

piControl-ERA and rewrite the relevant text "Fig. 1 (D) shows that the ERA-Interim circulation has an eastward displacement and the intensity measured by STRF is overestimated by 26% relative to ensemble piControl. There is a similar structure to the stream function differences between NCEP2 reanalysis and piControl, and the STRF is only overestimated by 3% relative to ensemble piControl."

(Line 316-317) We redraw the Fig.4 show the difference of ERA-piControl rather than piControl-ERA and rewrite the relevant text "The intensity anomalies relative to piControl from both the reanalysis data sets are less than 21% (Fig. 4)."

**Tropical atmospheric circulation response to the G1 sunshade geoengineering radiative forcing experiment**

Anboyu Guo[1], John C. Moore[1,2,3], Duoying Ji[1]

[1] College of Global Change and Earth System Science, Beijing Normal University, 19 Xinjiekou Wai St., Beijing, 100875, China

[2] Arctic Centre, University of Lapland, P.O. Box 122, 96101 Rovaniemi, Finland

[3] CAS Center for Excellence in Tibetan Plateau Earth Sciences, Beijing 100101, China

*Correspondence to:* John C. Moore (john.moore.bnu@gmail.com)

**Abstract.** We investigate the multi-Earth system model response of the Walker circulation and Hadley circulations under the idealized solar radiation management scenario (G1) and under abrupt4×$CO_2$. The Walker circulation multi-model ensemble mean shows changes in some regions but no significant change in intensity under G1, while it shows 4° eastward movement and $1.9 \times 10^9$ kg s$^{-1}$ intensity decrease in abrupt4×$CO_2$. Variation of the Walker circulation intensity has the same high correlation with sea surface temperature gradient between eastern and western Pacific under both G1 and abrupt4×$CO_2$. The Hadley circulation shows significant differences in behavior between G1 and abrupt4×$CO_2$ with intensity reductions in the seasonal maximum Northern and Southern cells under G1 correlated with equator-ward motion of the Inter Tropical Convergence Zone (ITCZ). Southern and Northern cells have significantly different response, especially under abrupt4×$CO_2$ when impacts on the Southern Ferrel cell are particular clear. The

Southern cell is about 3% stronger under abrupt4×$CO_2$ in July, August and September than under piControl, while the Northern is reduced by 2% in January, February and March. Both circulations are reduced under G1. There are significant relationships between Northern cell intensity and land temperatures, but not for the Southern cell. Changes in the meridional temperature gradients account for changes in Hadley intensity better than changes in static stability both in G1 and especially in abrupt4×$CO_2$. The difference in response of the zonal Walker circulation and the meridional Hadley circulations under the idealized forcings may be driven by the zonal symmetric relative cooling of the tropics under G1.

**1 Introduction**

The large-scale tropical atmospheric circulation may be partitioned into two independent orthogonal overturning convection cells, namely the Hadley circulation (HC) and the Walker circulation (WC), (Schwendike et al., 2014). The HC is the zonally symmetric meridional circulation with an ascending branch in the intertropical convergence zone (ITCZ) and a descending branch in the subtropical zone, and which plays a critical role in producing the tropical and subtropical climatic zones, especially deserts (Oort and Yienger, 1996). The WC is the asymmetric zonal circulation which extends across the entire tropical Pacific, characterized by an ascending center over the Maritime Continent and western Pacific, eastward moving air flow in the upper troposphere, a strong descending

center over the eastern Pacific and surface trade winds blowing counter to the upper winds along the equatorial Pacific completing the circulation (Bjerknes, 1969).

Observational evidence shows a poleward expansion of the HC in the past few decades (Hu et al., 2011) and an intensification of the HC in the boreal winter (Song and Zhang, 2007). Climate model simulations with increased greenhouse gas forcing also indicate a poleward expansion of the Hadley circulation,  (Hu et al., 2013; Ma and Xie, 2013; Kang and Lu, 2012; Davis et al., 2016). Vallis et al. (2015) analysed the response of 40 CMIP5 climate models finding that there was only modest model agreement on changes. Robust results were slight expansion and weakening of the winter cell HC in the Northern Hemisphere (NH). It is unclear how closely the model simulations match reality. Choi et al. (2014) and Quan et al. (2014) both suggest that reanalysis trends for the Hadley cell edges may be overstated, especially compared to independent observations, and model trends are in reasonable agreement with the reanalysis trends (Davis and Birner, 2017; Garfinkel et al., 2015), but choice of metric also matters (Solomon et al., 2016) when discussing trends.

Many authors have considered the impact of greenhouse gas forcing on the Hadley circulation, particular in respect of changes in the width of the tropical belt (e.g., (Frierson et al., 2007; Grise and Polvani, 2016; Johanson and Fu, 2009; Lu et al., 2007; Seidel et al., 2008), but far fewer have discussed changes in Hadley intensity (Seo et

al., 2014; He and Soden, 2015). The importance of tropical belt widening is of course due to its impact on the hydrological system, especially the locations of the deserts (Lau and Kim, 2015; Seager et al., 2010), which are a critically important for the habitability of several well-populated areas.

Observational evidence shows a strengthening and westward movement of the WC from 1979 to 2012 (Bayr et al., 2014; Ma and Zhou, 2016). However, the time required to robustly detect and attribute changes in the tropical Pacific WC could be 60 years or more (Tokinaga et al., 2012). Model results suggest a significant eastward movement with weakening intensity under greenhouse gas forcing (Bayr et al., 2014), and He and Soden (2015) propose that the sea surface temperature warming plays a crucial role in both the eastward shift and the weakening of WC. They also note that this weakening may be reversed by rapid land warming.

Geoengineering as a method of mitigating the deleterious effects anthropogenic climate change has been suggested as a complement to mitigation and adaptation efforts. For example, Shepherd et al. (2009) summarized the methodologies and governance implications as early as a decade ago. Solar radiation management (SRM) geoengineering can lessen the effect of global warming due to the increasing concentrations of greenhouse gases by reducing incoming solar radiation. This compensating of longwave radiative forcing with shortwave reductions necessarily leads to non-uniform effects around the globe, as summarized in results for many climate models in the Geoengineering Model Intercomparison Project (GeoMIP) by (Kravitz et al., 2013). This is due to the seasonal and diurnal patterns of short wave

forcing being far different from the almost constant long wave radiative absorption. In addition, SRM tends to produce net drying due to the decreasing in vertical temperature gradient as greenhouse gasses (GHGs) increase absorption in the troposphere while shortwave radiative forcing affects surface temperatures (Bala et al., 2011). These differences in short and long wave forcing impacts atmospheric circulation and hence precipitation patterns, summarized for the GeoMIP models by Tilmes et al. (2013). The general pattern of temperature change under abrupt4$\times$CO$_2$ includes accentuated Arctic warming, and least warming in the tropics. G1 largely reverses these changes, but leaves some residual warming in the polar regions and under-cools the tropics relative to piControl. SRM also reduces temperatures over land more than over oceans relative to abrupt4$\times$CO$_2$, and hence reduces the temperature difference between land and oceans by about 1°C. Extreme precipitation is affected by SRM such that heavy precipitation events become rarer while small and moderate events become more frequent (Tilmes et al., 2013). This is generally opposite to the impact of GHG forcing alone which tends to produce a "wet gets wetter and dry gets drier" pattern to global precipitation anomalies (Tilmes et al., 2013; Held and Soden, 2006). Finally tropical extreme cyclones have been shown to be affected by SRM in ways that do not simply reflect changes in tropical sea surface temperatures due to large scale planetary circulations and teleconnection patterns (Moore et al., 2015).

To date, few studies of the impact of SRM on tropical atmospheric circulation has been published. Ferraro et al. (2014) using an intermediate complexity climate model found tropical overturning circulation weakens in response to SRM with stratospheric sulfate aerosol injection. But SRM simulated as a simple reduction in total solar irradiance does not capture this effect. Davis et al. (2016) analyzed 9 GeoMIP models and report that the HC expands in response to a quadrupling of atmospheric carbon dioxide concentrations more or less proportionality to the climate sensitivity of the climate model, and shrinks in response to a reduction in solar constant. Smyth et al. (2017) report that decreases in  Hadley cell intensity drive the reduction in tropical precipitation under SRM, and that seasonal changes mean that the ITCZ has smaller amplitude northward shifts compared with no SRM.

The El Niño Southern Oscillation (ENSO) is the largest mode of multi-annual variability exhibited by the climate system in terms of its temperature variability and also for its socio-economic impacts. This tropical circulation pattern is intimately related to changes in the WC by their dependences on the Pacific Ocean zonal sea surface temperature gradient, and indirectly to the HC by its impacts on global energy balance. Few studies of climate model ENSO response to SRM have been made, with Gabriel and Robock (2015) finding that stratospheric aerosol injection by the GeoMIP G4 experiment produces no significant impacts on El Niño/Southern Oscillation. The

 SRM and GHG forcing in the G4 experiment are both relatively low compared with  the G1 experiment, since under G4 the GHG scenario is the modest RCP4.5 , which means that natural climate variability in the 50 year long period of SRM period may obscure features. 
[revised manuscript text omitted]
 HC intensity for the Southern cell as the average meridional stream-function between 900-100hPa over the area between 40°S and 15°N in July, August and September (JAS), and the Northern cell as the absolute value of mean meridional stream-function between 15°S and 40°N in January, February and March (JFM). We experimented with using narrower definitions of the Hadley cell (38°-15° or 35°-15°) in the 3 experiments, finding almost the same systematic offsets in intensities across the models and experiments. This is also true for each hemisphere separately. Departures in model ensemble mean intensity across the three experiments

for both hemispheres from an outer latitude of 40° range from 6.6-7% and 13.8-14% with outer latitudes of 38°and 35° respectively. So using the wide latitude bands we chose captures all the variability in the Hadley cells in all the models and experiments without introducing biases due to experiments or hemispheres. We use the 900 – 100 hPa levels (whereas typically 200 hPa has been the ceiling, (e.g., Nguyen et al., 2013)) to accommodate the raised tropopause under GHG forcing, while avoiding boundary effects.

**3 Walker circulation response**

**3.1 Intensity**

The annual mean state of zonal mass stream-function ($\psi_z$) calculated from 8 ensemble member mean piControl, ERA-Interim reanalysis and the NCEP2 reanalysis results are shown in Fig. 1. Zonal mass stream-function ($\psi_z$) can intuitively depict the WC which exhibits its strongest convection (positive values) in the equatorial zone across the Pacific. The WC center is around 500hPa and 160°W. Fig. 1 (D) shows that the ERA-Interim circulation has an eastward displacement and the intensity measured by STRF is overestimated by 26% relative to ensemble piControl. There is a similar structure to the stream function differences between  NCEP2 reanalysis and piControl, and the STRF is only overestimated by 3% relative to ensemble piControl.

The relative changes from piControl under G1 and abrupt4×CO₂ experiments are

shown in Fig. 2. The features of WC are very similar in both the G1 and piControl experiments shown in Fig. 2 (A). In abrupt4×CO$_2$ differences are larger, and include a rise in vertical extent of the circulation and an eastward shift in Fig. 2 (B). This is quantifiably confirmed by the STRF index increase of just 0.3% in G1 but a significant decrease of 7% in abrupt4×CO$_2$ relative to piControl, (Table 2). However, only 5 out of 8 models agree on the sign of the changes in abrupt4×CO$_2$ and there is much diversity between individual models (Fig. S3).

**3.2 Position**

The vertically averaged zonal mass stream-function ($\psi_z$) for the ensemble means of the 3 experiments as a function of longitude are shown in Fig. 3. To quantitatively measure the position change of the WC we use the western edge index. The ERA-Interim and NCEP2 reanalysis data respectively show 10.5° and 18° more easterly positions than the piControl state. The WC shifts 0.5° westward in G1 and 4° eastward in abrupt4×CO$_2$ relative to piControl for the multi-model ensemble mean. There is significant change in the ensemble mean position and strength under abrupt4×CO$_2$, but not G1 in Table 2. However  only 5 out of 8 models agree on the sign of the changes, so the inter-model differences are rather large in this case. In the G1 experiment, the WC strengthens over the western Pacific around 130°E to 150°E and

weakens over the eastern Pacific around 115°W to 80°W, indicating a westward movement relative to piControl, (Table 2). Thus the pattern is the opposite of that seen under abrupt4×CO$_2$ in Fig. 3 (B).

Under G1 there is a westward shift in the ascending branch of the circulation from about 30°E to about 20°E as indicated by comparing the red shaded region around 30°E in Fig. 2 (A) with the piControl result in Fig. 1 (C). Fig. S3 shows the anomaly is present in CanESM2, CCSM4, and NorESM1-M, while 3 models show almost no change (and indeed are missing the African features in their piControl simulation). BNU-ESM shows the opposite anomaly while GISS-E2-R shows a complex pattern. There is only small change in the STRF zero crossing location in the region (Fig. 3 (B)) because of the anomalies are not vertical. This position is at the transition from tropical West African rainforest to wood and grassland in East Africa under present climates. The movement westward would impact the rain forests of the Congo basin. There is no similar positional change under abrupt4×CO$_2$ in the region, though there are many more changes in the circulation as a whole.

**4 Hadley circulation intensity response**

The climatology of the meridional mass stream-function ($\psi_m$) calculated from multi-model ensemble mean are shown in Figs. 4 and 5 and the individual models are shown in Fig. S4. This can naturally describe the HC with a clockwise rotation in the NH and an anticlockwise rotation in the Southern Hemisphere (SH). The Southern Hadley cell

width spans nearly 35°of latitude and the Northern Hadley cell about 25° latitude. The intensity anomalies relative to piControl from both the reanalysis data sets are less than 21% (Fig. 4).

Circulation anomalies under abrupt4×$CO_2$ (Fig. 5, (B)), show enhanced overturning aloft and weakened overturning at lower levels in both Northern and Southern Hadley cells. The elevation of the circulation upper branches rises with increased GHG concentration, as previously noted (Vallis et al., 2015), and is likely a consequence of the rise in tropopause height due to GHGs. The Southern cell shows a complex anomaly structure with positive anomaly between 45°S-65°S also in the Ferrel cell circulation that borders it at higher southern latitudes. The Northern cell anomaly is simpler in comparison. Under G1 the changes (Fig. 5(A)) are largest near the equatorial margins of the cells, with a clear increase in the strength of the ascending current. There is no significant change in the upper branch of the circulation showing that the tropopause is returned to close to piControl conditions despite the greenhouse concentrations being raised. Seasonal differences illustrate the changes induced under the experiments in a clearer way than the annual ensemble result (Fig. 6).

In JAS, when the ITCZ is located furthest north around 15°N, the G1 anomaly indicates a reduction in the upward branch of the Southern cell, or equivalently, a southern migration of the ITCZ. Similarly in JFM there is a corresponding reduction in strength of the upwelling branch of the Northern cell (Fig. 6. (C) and (D)).

This is a similar result as obtained by Smyth et al. (2017) who considered the ITCZ position to be defined as the centroid of precipitation, and found changes in position of fractions of a degree. Fig. 7 (B) shows that  the modelled motion of the ITCZ explains 73% of the variance in intensity of the JAS Southern cell peak intensity , which is significant at the 95% level. Thus the larger the model reduction in intensity the more the boundary of the ITCZ moves equatorward. The correlation for the JFM Northern cell (Fig. 7 (A))is not strong to be significant though still indicates correlation between intensity and ITCZ position changes. The combined seasonal effect of both cell changes is a reduced migration of the upwelling branches of the circulation cells across the equator, as was also noted by Smyth et al. (2017).

The GISS-E2-R model has strikingly different anomalies under both G1 and abrupt4×$CO_2$ compared with other models, with much more variability and more changes in sign of rotation not only within the Hadley cell but in the surrounding Ferrel cells. If we exclude this model from the ensemble, we get an even clearer result showing that the movement of the equatorial edge of the Hadley cells (the ITCZ) totally dominates the response under G1 (Fig. S5).

The situation under abrupt4×$CO_2$ is more complex (Fig.

 (E) and (F)). The expansion of the tropics has been noted both in GHG simulations and observationally (Davis et al., 2016; Hu et al., 2011), along with the larger southern expansion. The extratropical changes in the Ferrel circulation are also more pronounced in the SH.

Reduction in strength of the NH winter cell was also a robust result of climate models under RCP8.5, while, the Southern cell exhibited almost no change (Vallis et al., 2015). Our results in Fig. 8 show that the multi-model ensemble mean  reduction Hadley intensity under G1 of $-18 \times 10^8$ kg s$^{-1}$ and of $-7 \times 10^8$ kg s$^{-1}$ for abrupt4$\times$CO$_2$. The JAS Southern Hadley intensity  exhibits a fall of $-16 \times 10^8$ kg s$^{-1}$ under G1 but an increase of $23 \times 10^8$ kg s$^{-1}$ under abrupt4$\times$CO$_2$. At least 6 out of 8 models agree on these sign of changes in both hemispheres and scenarios. Thus the SH results differ for abrupt4$\times$CO$_2$ from those presented in Vallis et al. (2015). The anomalies for most models are significant, and the ensemble means are 8 standard errors from zero and thus very highly significant.

**5 ENSO variability of Walker and Hadley circulations**

Many previous study have concluded that the WC weakens and shifts eastward during El Niño, with opposite effects under La Niña, (Ma and Zhou, 2016; Power and Kociuba, 2011; Yu et al., 2012; Power and Smith, 2007). HC shrinks and strengthens during El Niño events, while expanding and weakening during La Niña, (Nguyen et al., 2013; Stachnik and Schumacher, 2011). The G1 solar dimming SRM impacts on the Walker and HC during ENSO events will be discussed in this section.

The WC difference between G1, abrupt4$\times$CO$_2$ and piControl vary among models during ENSO events (Fig. S6). But the multi-model ensemble mean presents a clear picture (Fig. 9). The result show that features of WC response to ENSO are significantly changed under abrupt4$\times$CO$_2$ compared with piControl, while G1 compares quite closely to piControl. Differences between G1 and piControl only manifest themselves at the eastern (about 165°E-180°E) and western (about 120°W-90°W) sides of WC, with a significant westward movement during El Niño, and no significant changes during La Niña.

In contrast under abrupt4$\times$CO$_2$ almost the whole WC (about 165°E-105°W) strengthens in intensity and the western edge shifts westward at the 95%

statistical significance level during El Niño relative to piControl. During La Niña there is a significant eastward movement in general.

HC responses to ENSO under G1, abrupt4×CO$_2$ and piControl vary among models (Fig. S7). Fig. 10 shows the ensemble mean results. As with the WC, the climatological features of the Hadley cell show more significant changes under abrupt4×CO$_2$ than G1 compared with piControl.

The most notable feature of Fig. 10 is the increase in intensity during La Niña between 10°S and 10°N under abrupt4×CO$_2$. This corresponds to changes in the Southern Hadley cell (remembering that the axis of the Hadley cells is northwards of the equator). Also under the same conditions there is weakening of the Northern Hadley cell between 10°and 20°N. The same features are almost as noticeable for abrupt4×CO$_2$ for El Niño conditions and hence is a general feature of the abrupt4×CO$_2$ climate state. Beyond the Hadley cells there are modest, but statistically significant changes in the Ferrel circulations, particularly in the SH. Changes under G1 in comparison are much smaller than under abrupt4×CO$_2$, though there are significant reductions in intensity near the margins of the Hadley cells. The Northern cell is more affected in El Niño, while the Southern one more in La Niña states.

**6 Hadley and Walker circulations relationships with temperature**

**6.1 Walker Circulation**

Changes in tropical Pacific SST dominate the global warming response of the

WC change (Sandeep et al., 2014). A reduced SST gradient between eastern and western Pacific drives the weakening of WC that was seen in a quadrupled $CO_2$ experiment (Knutson and Manabe, 1995). The temperature difference between eastern and western Pacific, $\triangle$SST, explains 96% of the inter-model variance in the strength of the WC in the G1-piControl anomalies and 79% of the variance for abrupt4×$CO_2$-piControl, (Fig. 11). There is no difference in model behavior between the G1 and abrupt4×$CO_2$ anomalies and $\triangle$SST explains 83% of the overall variance. Despite a temperature transient of at a decade or so (e.g. Kravitz et al., 2013) in the abrupt4×$CO_2$ simulation and the lack of any transient in STRF (Fig. S1), the relationship with $\Delta$SST is nearly as good as for piControl. This suggests that there is no difference in mode of behavior of the WC under solar dimming SRM or GHG forcing, in contrast with the changes seen in the Hadley cells.

 The correlation between yearly STRF and global 2 m temperatures are shown in Fig. 12 and the individual models are shown in Fig. S8. We discard first 20 years for G1 and abrupt4×$CO_2$ to remove the temperature transients. In G1 all models except CanESM2 and MIROC-ESM have strong negative correlations between STRF and tropical Pacific temperatures. BNU-ESM, CCSM4 and NorESM1-M show a positive correlation with temperatures in the South Pacific convergence zone

(SPCZ) and its linear extension in the South Atlantic. These features are generally muted or absent in the piControl simulations. Experiments with an atmospheric circulation model (Van der Wiel et al., 2016) suggest that a key feature of the diagonal structure of the SPCZ is the zonal temperature gradient in the Pacific which allows warm moist air from the equator into the SPCZ region. This moisture then intensifies (diagonal) bands of convection carried by Rossby waves (Van der Wiel et al., 2016). Two of the three models with  positive correlation between STRF and SPCZ temperatures , CCSM4 and NorESM1-M, have increased STRF and ΔSST under G1 (Fig. 11) suggesting that this mechanism is responsive in at least some of the models to G1 changes in forcing. The SPCZ is the only part of the ITCZ that extends beyond the tropics and so may be expected to be more subject to the meridional gradients in radiative forcing produced by G1. The correlations under abrupt4×$CO_2$ are more variable across the models, though some of models like IPSL-CM5A-LR, MIROC-ESM and HadGEM2-ES exhibit widespread anti-correlation between STRF and temperatures; the spatial variability suggests that this not due to the strong transient response in global temperature rises under abrupt4×$CO_2$.

**6.2 Hadley Circulation**

We now consider how surface temperature changes may impact the HC. To remove the transients, we only use the last 30 years for G1 and abrupt4×$CO_2$. The decrease of the Northern Hadley cell intensity in JFM (Fig. 8) correlates with Northern hemispheric land temperatures (Fig. 13),

explaining 58% of the variance in model anomaly under G1 – which is nevertheless not significant at the 95% level - and 81% under abrupt4×$CO_2$. NH land temperature also explains 83% of the G1 anomaly in the Southern Hadley cell in JAS, but has no impact on the abrupt4×$CO_2$ anomaly. Both SRM and GHG forcing modifies the land-ocean temperature difference relative to piControl and so conceivably affects HC, for example by changing the hemispheric temperature and the position of the ITCZ (Broccoli et al., 2006). Under abrupt4×$CO_2$ land-ocean temperature differences in the tropics (between 30° N and 30°S) are reduced to essentially zero, while under G1 differences in the tropics are 1.2°C which is not significantly different from the piControl difference of 1.4°C. Since the largest continental land masses are in the NH, we would expect any differences in HC induced by land-ocean contrasts in the NH to be visible in the Southern Hadley cell. We explored the impact of land-ocean temperature differences by considering differences in the surface temperatures over Tibet and the whole tropical ocean temperature  (Fig. S9). Results were similar as for Fig. 13, with significant correlations for G1 in the Southern Hadley cell.

Seo et al. (2014) examine the relative importance of changes in meridional temperature gradients in potential temperature, subtropical tropopause height, and static stability on the strength of the HC. They find that according to both scaling theory based on the Held and Hou (1980) and the Held (2000) models, and analysis of 30 CMIP5 models forced by the RCP8.5 scenario, that it is the meridional temperature gradient that is the most important factor.

We used the same procedure as Seo et al. (2014) on the 4 models (BNU-ESM, IPSL-CM5A-LR, HadGEM2-ES, MIROC-ESM) that provide all the fields needed under G1 and abrupt4×$CO_2$ scenarios (Table 3). The changes in ensemble mean circulation intensity are similar under G1 and abrupt4×$CO_2$, as are the changes in potential temperature gradients relative to piControl, but the changes in static stability are very different between the experiments. The tropospheric heights also change between G1 and abrupt4×$CO_2$ scenarios, with small reductions under G1 and about a 3% and 0.9% increase respectively in Southern and Northern cells under abrupt4×$CO_2$. We used the two scaling relations given by Seo et al., (2014) to also estimate the change in Hadley intensity based on the changes in temperature gradients, static stability and tropospheric height for the ensemble mean of the 4 models (Table 3). Both formulations give fairly similar numbers for the estimated change in Hadley intensities in Northern and Southern cells under G1 and abrupt4×$CO_2$. These estimates agree with the simulated changes in intensities under G1, but are very different from those simulated under abrupt4×$CO_2$. The obvious cause of the discrepancies under abrupt4×$CO_2$ is the change in static stability, which in both model scaling formulations leads to 18-25% reductions in Hadley intensity compared with the ensemble model simulated changes of about ±4%. This supports the analysis of Seo et al. (2014) that it is the meridional temperature gradient that is the dominant factor in determining the strength of the HC.

**7 Discussion**

Our main purpose in this study has been to analyze the response of Walker and Hadley circulation to greenhouse gas and solar dimming geoengineering forcing simulated by abrupt4×CO$_2$ and G1 experiments. A clear Walker circulation westward movement during El Niño and an eastward movement during La Niña are shown nearly everywhere along the equator in abrupt4×CO$_2$ relative to piControl. However only the eastern and western side of Walker circulation manifest the same movement during ENSO events in G1 relative to piControl. The range and amplitudes of significant changes are smaller in G1 than in abrupt4×CO$_2$. We note a potentially important change in position of the walker Circulation associated with the West African rainforest and East African grassland zones, under G1, with potential for the encroachment of a drier climate into the Congo basin.

Davis et al., (2016) note an expansion in the Hadley cells in proportion to the temperature rises in the models under both G1 and abrupt4×CO$_2$. Here, weHe and Soden (2015) conclude from experiments designed to elucidate the role of various forcings on tropical circulation that weakening of the WC under GHG forcing is primarily due to mean SST warming. They also note that increased land-sea temperature contrast results in strengthening of the circulation, and also that while the pattern of GHG warming is close to an El Niño, there are sufficient differences to produce quite different responses in the WC. We may therefore expect that changes under G1 compared with pure GHG forcing would manifest themselves given the changes in both the direct and indirect CO$_2$ forcings. What we observe though is that changes in the WC are modest, and examination of the dependence on intensity as a

function of zonal Pacific Ocean temperature differences (Fig. 11) show no differences between the GHG and G1 forcings. Similarly we find no change in the intensity with land-ocean temperature gradients.

We see large changes throughout the whole Hadley cell circulation under abrupt4×$CO_2$. We also see that the northern boundary of the Southern cell tends to expand even further northwards with a corresponding weakening of the northernNorthern cell during La Niña conditions. Global temperatures are relatively reduced during La Niña years. Beyond the Hadley cells there are modest, but statistically significant changes, particularly in the SH Ferrel circulations, particularly in the Southern hemisphere with poleward movement. Changes under G1 in comparison are much smaller than under abrupt4×$CO_2$, though there are significant reductions in intensity near the margins of the Hadley cells and these are related to equator-ward motion of the ITCZ. The northernNorthern cell is affected more in El Niño, while the southernSouthern one more by La Niña states.

Davis et al. (2016) show that southernSouthern Hadley cell expansion in the tropics is on average twice the northernNorthern Hadley expansion. The idealized forcings in abrupt4×$CO_2$ and G1 show this cannot be due to stratosphere ozone depletion – the mechanism sometimes used to account for the similar observed greater expansion of the southernSouthern Hadley cell (Waugh et al., 2015). The changes in width of the tropical belt is strongly dependent on the tropical static stability in the models according to the Held and Hou (1980) scaling, that is with the potential temperatures at the tropical tropopause (100 hPa) and the surface. Since the adiabatic lapse rates scales with surface

temperature, this is also reflected in the surface temperature. Consideration of simplified convective systems based on moist static energy fluxes (Davis, 2017), or by making some assumptions with the Held (2000) and Held and Hou (1980) models led Seo et al. (2014) to suggest Hadley cell intensity scales according to the equator-pole temperature gradient.

Furthermore the intensity of the HC is expected to decrease as it expands and also in response to an accelerated hydrological cycle. An enhanced hydrological cycle is expected under GHG forcing, but not SRM which leads to net drying (Kravitz et al., 2013). This is cannot be a complete explanation for circulation changes since the HC also depends on the evolution of the baroclinic instabilities in the extratropics, which may have quite different response to climate warming (e.g. Vallis et al., 2015). Our analysis of intensity shows differences in behavior between Southern and Northern cells, and in particular a lack of a strong dependences on temperature gradients for the Southern cell. The difference in behavior between Northern and Southern Hadley cells has not been explained to date. Seo et al. (2014) note that under RCP8.5 forcing, models of the Southern Hadley cell changes are split almost equally between those predicting increases in intensity and those that suggest decreases, whereas all but 1 of 30 models predicts a decrease in the Northern cell. We note that the robustly understood vertical expansion of the circulation as the tropopause rises under abrupt4×CO$_2$, has been associated with a decrease in the circulation

intensity (Seo et al., 2014; He and Soden, 2015) in climate models forced by GHGs, and as expected from considerations of Clausius-Clapeyron scaling if relative humidity is relatively constant, as summarized by Vallis et al. (2015). This is not the case for the scaling functions from Seo et al., (2014; Table3), where tropopause height change is proportional to intensity change. Nor it is consistent with increases simulated in the Southern Hadley cell intensity and simultaneous decreases in the Northern one relative to piControl, although both are stronger than under the G1 forcing. Our analysis of the relative importance of factors in driving intensity suggests, as with Seo et al (2014), that the meridional temperature gradient plays the dominant role rather than tropopause height or static stability changes.

Grise and Polvani (2016) explored how the dynamic response of the atmosphere, including metrics such as Hadley cell edge, varied with model climate sensitivity, that is the mean temperature rise associated with doubled $CO_2$. They found significant correlation across a suite of CMIP5 models running the abrupt4$\times CO_2$ were largely confined to the SH, and also that the pole-to-equator surface temperature gradient accounted for significant parts of the dynamic variability that was not dependent on the mean temperature. However, we find that the response times of the HCs to changes in radiative forcing are very fast, as shown by the lack of transients in the simulated time series. Sea surface temperatures, especially under the strong abrupt4$\times CO_2$ forcing takes at least a decade and parts of the system, such as the deeper ocean , would require even longer to reach equilibrium. Under abrupt4$\times CO_2$ the global land-ocean temperature difference is

reduced by about 1.3°C relative to piControl, while G1 reduces the contrast by only 0.3°C. The NH continents have faster response times than the oceans and so we would expect the SH to be much further from an equilibrium response than the Northern. This is also reflected in the lack of an equivalent to the "Arctic amplification" seen in the NH under both observed and simulated forcing by GHGs. The lack of anomalous Southern polar warming is linked to the much cooler surface temperatures in the Antarctic mitigating against both temperature feedbacks and the ice-albedo feedback mechanism (Pithan and Mauritsen, 2014). The speed of response of the circulation changes calls into question the importance of static stability and meridional gradients in driving the changes in the circulation, since the circulation responds faster. Bony et al. (2013) attributed rapid changes in circulation in quadrupled $CO_2$ as due to direct $CO_2$ forcing. Fast response could also be a result of cloud feedback, land-ocean temperature differences and perhaps humidity, which are also important for poleward energy transport in G1 (Russotto and Ackerman, 2018; Russotto and Ackerman, in review ACP). Low cloud fraction decrease under G1, warming the planet by reducing the reflection of solar shortwave radiation, but atmospheric humidity is reduced allowing heat to escape, and less energy is transported from tropics to poles.

Our analysis of circulation intensity changes and their dependence on temperature changes shows quite different sets of behavior under G1 than under abrupt4×$CO_2$ for the Hadley but not the WC. The response under G1 relative to piControl is a slight overcooling of the tropics relative to the global mean temperature

(Kravitz et al., 2013). Experiments with idealized climate models (Tandon et al., 2013) show that heating at the equator alone tends to reduce the Hadley cell width, while wider heating in an annulus around the outer tropics (20°-35°) tends to produce a complex response to circulation in both Hadley and Ferrel cells, more reminiscent of the anomaly patterns seen under abrupt4×CO$_2$. The climate forcing under G1 is designed to be zonally symmetric, and that may explain lack of impact in the WC under both G1 and GHG forcing. There are clear changes of Hadley cells under the latitudinal varying forcing of G1 . The reduction in incoming shortwave radiation in G1 would intuitively mean reduced heating, sea surface temperatures and moisture flux in the ITCZ, which follows the movement of the sun.  Analysis of extreme precipitation events in daily data from the GeoMIP models (Ji et al., submitted to ACP) shows that the annual wettest consecutive five days are drier under G1 along a seasonal path that follows the ITCZ motion, while precipitation extremes increase in the tropical dry seasons. This result is consistent with the variation in the Hadley intensity cell seen here.

**8 Summary**

Our main purpose in this study has been to answer the following questions: Does the G1 scenario counteract position and intensity variations in the Walker and HCs caused by the GHG long wave forcing under abrupt4×CO$_2$? How does the tropical

atmospheric circulation, including the Walker and HCs, respond to warm and cold phases of the El Niño Southern Oscillation (ENSO) in G1 and abrupt4×$CO_2$?

The WC in G1 displays insignificant increases in intensity and no shift in its western edge in the Pacific Ocean relative to piControl and hence does counteract the changes from GHG forcing. There is a potentially important change in position of the WC associated with the West African rainforest and East African grassland zones under G1, with potential for the encroachment of a drier climate into the Congo basin. In contrast, the HC shows larger changes under G1 that are not simple reversals of those induced by GHG forcing on piControl climate. There are asymmetric responses between the hemispheres under both GHG and solar dimming that are correlated with direct forcings rather than adjustment of sea surface temperatures, and correlated with changes in meridional and land-ocean temperature gradients. These differences in response of the Hadley and Walker circulations are consistent with the zonally invariant forcing of both solar dimming and GHGs and the meridionally varying solar dimming.

A clear WC westward movement during El Niño and an eastward movement during La Niña are shown nearly everywhere along the equator in abrupt4×$CO_2$. However the eastern and western boundaries of the WC shift westward during El Niño in G1 relative to piControl. The range and amplitudes of significant changes are smaller in G1 than in abrupt4×$CO_2$. The same is true in general for the Hadley cell. Under abrupt4×$CO_2$ the Northern Hadley cell significantly decreases in intensity under both la Niña and El Niño conditions while under G1 the decreases are smaller and limited to each cell's poleward boundaries.

Both models and the limited observational data available on the HC indicate that it is not zonally symmetric: there are intense regions of circulation at the eastern sides of the oceanic basins (Karnauskas and Ummenhofer, 2014), while elsewhere circulation is reversed, and much of the natural variability of the circulation is related to ENSO (Amaya et al., 2017). This and the opposite correlations with surface temperatures in the Pacific and SPCZ with STRF under G1 (Fig. 12) suggests an interplay between HC and WC that could repay further consideration of model data at seasonal scales. The importance of the tropical ocean basins as genesis regions for intense storms also suggests that changed radiative forcing there under SRM could cause important differences in seasonal precipitation extremes, that may be hidden in monthly or annual datasets.

*Acknowledgements.* We thank two anonymous referees for very constructive comments, the climate modeling groups for participating in the Geoengineering Model Intercomparison Project and their model development teams; the CLIVAR/WCRP Working Group on Coupled Modeling for endorsing the GeoMIP; and the scientists managing the earth system grid data nodes who have assisted with making GeoMIP output available. This research was funded by the National Basic Research Program of China (Grant 2015CB953600).

FIGURES

[Figure]

[Figure]

**Figure 1.** Walker circulation in the ERA-Interim reanalysis (A), NCEP2 reanalysis (B), model ensemble mean  under piControl (C) and difference between ERA-Interim and piControl (D). Color bar indicates the value of averaged zonal mass stream-function ($10^{10}$ kg s$^{-1}$). Warm color (positive values) indicate a clockwise rotation and cold color (negative values) indicate an anticlockwise rotation.

[Figure]

**Figure 2.**  Shading indicates model ensemble mean zonal stream-function anomalies  ($10^{10}$kg s⁻¹) G1-piControl (A) and abrupt4×$CO_2$

[Figure]

[Figure]

**Figure 4.**  (B). Warm colors (positive values) indicate a clockwise rotation and cold colors (negative values) indicate an anticlockwise rotation. Contours indicate the value of averaged meridional mass stream-function ($10^{10}$kg s$^{-1}$) in piControl as plotted in Fig. 1 (C).

[Figure]

[Figure]

**Figure 3.** The vertically averaged zonal mass stream-function ($10^{10}$ kg s$^{-1}$) (A) in piControl, G1, abrupt4×CO$_2$ experiment for ensemble mean , ERA-Interim and NCEP2. And their difference relative to piControl (B).

[Figure]

**Figure 4.** Hadley circulation in the ERA-Interim reanalysis (A), NCEP2 reanalysis (B), model ensemble mean under piControl (C) and difference between ERA-Interim and piControl (D). Color bar indicates the value of averaged meridional mass stream-function ($10^{10}$ kg s$^{-1}$). Warm color (positive values) indicate a clockwise rotation and cold color (negative values) indicate an anticlockwise rotation.

[Figure]

**Figure 5.** Shading indicates model ensemble mean zonal stream-function anomalies $(10^{10} kg\ s^{-1})$ G1-piControl (A) and abrupt4×$CO_2$-piControl (B). Warm colors (positive values) indicate a clockwise rotation and cold colors (negative values) indicate an anticlockwise rotation. Contours  indicate the value of averaged meridional mass stream-function $(10^{10} kg\ s^{-1})$ in piControl as plotted in Fig. 4 (C).

[Figure]

**Figure 6.** Model ensemble mean meridional stream-function in

piControl (A) and (B), anomalies relative to piControl for G1 (C) and (D) and anomalies relative to piControl for abrupt4×$CO_2$ experiments (E) and (F). (A), (C) and (E) indicate JAS months, (B), (D) and (F) indicate JFM months. Color bar indicates the value of averaged meridional mass stream-function ($10^{10}$ kg s$^{-1}$). Warm colors (positive values) indicate a clockwise rotation and cold colors (negative values) indicate an anticlockwise rotation. Contour indicate the value of averaged meridional mass stream-function ($10^{10}$kg s$^{-1}$) in piControl.

[Figure]

[Figure]

**Figure 7.** Change of Hadley cell intensity as a function of ITCZ position under G1 relative to piControl across the models. for the Northern Hadley cell in JFM (A) and the Southern Hadley cell in JAS (B). The ITCZ position is defined from the centroid of precipitation (Smyth et al., 2017).

[Figure]

**Figure 8.** Anomalies ($10^{10}$ kg s$^{-1}$) relative to piControl amongst models in Hadley circulation for the Southern cell in JAS (left panel), defined as the magnitude of the mean meridional stream-function between 15°N and 40°S, and (right panel) the Northern cell in JFM, defined as the magnitude of the mean meridional stream-function between 15°S and 40°N. The dot size for the models is about 1 standard error of the model mean.

[Figure]

**Figure 9.** The vertically averaged of zonal mass stream-function under ENSO. For El Niño or La Niña conditions, blue line in each panel represent the vertically averaged of zonal mass stream-function ($10^{10}$ kg s$^{-1}$) under piControl. Red line in top row is G1 and bottom row abrupt4×CO$_2$. Thick lines denote locations where circulation changes are significant at the 95% confidence level. The 16%-84% range across the 8 individual models are show by light blue shading.

[Figure]

**Figure 10.** The vertically averaged of meridional mass stream-function under ENSO. For El Niño or La Niña conditions, blue line in each panel represent the vertically averaged of zonal mass stream-function ($10^{10}$ kg s$^{-1}$) under piControl. Red line in top row is G1 and bottom row abrupt4×CO$_2$. Thick lines denote locations where circulation changes are significant at the 95% confidence level. The 16%-84% range across the 8 individual models are show by light blue shading.

[Figure]

**Figure 11.** Model mean monthly anomalies relative to each model's piControl of STRF and ΔSST. Positive value of STRF and ΔSST indicate strengthening of the Walker circulation.

[Figure]

piControl

G1

4xCO2

Longitude (°)

Latitude (°)

-0.8 -0.7 -0.6 -0.5 -0.4 -0.3 -0.2 -0.1 0 0.1 0.2 0.3 0.4 0.5 0.6 0.7 0.8

[Figure]

**Figure 12.** Mean correlation between yearly STRF and global gridded 2 m temperatures for 100 years of piControl (A), and the final 30 years of G1 (B) and abrupt4×CO2 (C) experiments for 8 models ensemble mean.

[Figure]

**Figure 13.** Hadley intensity mean model anomalies versus the Northern hemisphere land temperature for the Northern Hadley cell  in JFM (A) and the Southern Hadley cell in JAS (B). Positive value of Hadley intensity indicates Hadley circulation strengthening regardless of the direction.

**Table 1.** The GeoMIP, CMIP5 models and reanalysis data used in the paper

[revised manuscript text omitted]

[1] The Hadley intensity $\psi_m$ is described in section 2.3 and we use JFM in the Northern hemisphere and JAS in the Southern hemisphere.

---

## Referee Report (RR1)

**Guo** *et al.* **2018**

**acp-2018-141**

The authors have addressed all the suggestions, substantially improving the clarity of the paper. I think the manuscript should be accepted for publication.